# Generation and trapping of a mesoderm biased state of human pluripotency

Dylan Stavish [1✉], Charlotta Böiers [2], Christopher Price [1], Thomas J. R. Frith [1], Jason Halliwell [1], Ingrid Saldaña-Guerrero [1], Jason Wray[2], John Brown [2], Jonathon Carr[1], Chela James[2], Ivana Barbaric[1], Peter W. Andrews [1✉] & Tariq Enver[2]

We postulate that exit from pluripotency involves intermediates that retain pluripotency while simultaneously exhibiting lineage-bias. Using a *MIXL1* reporter, we explore mesoderm lineage-bias within the human pluripotent stem cell compartment. We identify a substate, which at the single cell level coexpresses pluripotent and mesodermal gene expression programmes. Functionally these cells initiate stem cell cultures and exhibit mesodermal bias in differentiation assays. By promoting mesodermal identity through manipulation of WNT signalling while preventing exit from pluripotency using lysophosphatidic acid, we 'trap' and maintain cells in a lineage-biased stem cell state through multiple passages. These cells correspond to a normal state on the differentiation trajectory, the plasticity of which is evidenced by their reacquisition of an unbiased state upon removal of differentiation cues. The use of 'cross-antagonistic' signalling to trap pluripotent stem cell intermediates with different lineage-bias may have general applicability in the efficient production of cells for regenerative medicine.

[1] The Centre for Stem Cell Biology, Department of Biomedical Science, University of Sheffield, Western Bank, Sheffield S10 2TN, UK. [2] Stem Cell Laboratory, Department of Cancer Biology, University College London Cancer Institute, 72 Huntley St, London WC1E 6AG, UK. ✉email: d.stavish@sheffield.ac.uk; p.w.andrews@sheffield.ac.uk

Research of pluripotent stem cells (PSC), whether embryonic stem cells or induced pluripotent stem (IPS) cells, has often centred on robustly maintaining pluripotency. Less well investigated is how cells leave the pluripotent stem cell compartment. This is central for a biological understanding of how differentiation is regulated and lineage is specified. Since the use of pluripotent cells in regenerative medicine requires the efficient production of differentiated derivatives, delineating the cellular trajectories by which cells transit from pluripotency into lineage commitment is key. The existence of developmental intermediates on this trajectory may, in part, account for the cellular heterogeneity exhibited by in vitro cultures of human PSC[1].

Heterogeneity has been described in respect of: (i) Subsets of cells with different self renewal potential[2,3] identified based on surface antigen expression; and (ii) subsets with differential expression of lineage affiliated genes have been described; critically single cell studies have shown these to coexist with stem cell programmes[4–6]. Importantly, experimental assays of differentiation have demonstrated that these patterns of gene expression reflect interconvertible substates that functionally encode differential lineage bias within the stem cell compartment[4,5,7]. Reporter gene strategies have been particularly useful in this regard. We recently showed, using a fluorescent reporter, that a substate of human PSC expressed the early endoderm related marker *GATA6*[4]. *GATA6* positive cells were able to regenerate long-term pluripotent cultures yet their spontaneous differentiation favoured endodermal lineages. That study supports the notion—at least for endoderm—that these substates, which coexpress signatures of pluripotency and differentiation represent differentiation intermediates, may exist normally as transient states. Whether these exist for other lineages is not yet known. It is noteworthy that the pluripotent state of human PSC is itself also transient in development and is in effect trapped when human PSC are cultured in vitro. Recently, culture systems that aim to trap cells in an earlier naïve stage of development has been reported[8–10], but whether cells can be trapped further along a particular differentiation trajectory has been less well explored. Cross-antagonism of pro-pluripotency and pro-differentiation signalling in a controlled environment could potentially provide a strategy allowing for the propagation of a pluripotent intermediate with a specific lineage bias.

Herein we use a *MIXL1* reporter[11] to delineate the cellular trajectory from pluripotency to committed mesoderm progenitors. On this trajectory, we identify within the stem cell compartment a developmental intermediate that also exhibits mesoderm lineage bias. Using a cross antagonism strategy, we can trap and expand this intermediate while retaining its capacity to revert to an unbiased stem cell state. The identification and characterisation of mesodermal biased pluripotent intermediates informs our understanding of how mesoderm lineage specification occurs and could provide an attractive starting point for directed differentiation towards mesodermal derivatives.

## Results

### Identifying substates expressing *MIXL1*-GFP in feeder culture.
The expression of *MIXL1* in early differentiation of human PSC towards mesodermal derivatives[12] identified it as a candidate gene to assess for early lineage biased substates. We utilised the *MIXL1* reporter line engineered by Davis et al.[11], which has an enhanced GFP sequence inserted into the first exon of one of the *MIXL1* alleles of the HES3 human PSC line. This line has been used previously by many others investigating differentiation of human PSC, without any overt effects of the reporter's presence[11–15]. Throughout this paper, we have used expression of GFP as a measure of the *MIXL1* transcriptional state, which we refer to

throughout the manuscript as *MIXL1*-GFP. When cultures of the *MIXL1* reporter (HES3-*MIXL1*) were grown on mouse embryo fibroblast (MEF) feeder cells in medium containing Knock-out Serum Replacement (MEF/KOSR condition), they typically contained a subpopulation of 5–20% of cells that co-expressed *MIXL1* and SSEA-3, a surface antigen that we have previously used as a sensitive marker of undifferentiated stem cells[3,16,17] (Fig. 1a).

To examine the relationship of these *MIXL1*-GFP(+)/SSEA-3 (+) cells to the other subpopulations of *MIXL1*-GFP(−)/SSEA-3 (+) and *MIXL1*-GFP(+)/SSEA-3(−) cells, putatively stem and differentiated cells, respectively, and to test whether they co-express other markers that are separately indicators of a differentiated or undifferentiated state, we carried out RNA sequencing (RNAseq) analysis of these three subpopulations, isolated by fluorescence-activated cell sorting (FACS). The expression of *MIXL1*-GFP correlated well with the *MIXL1* gene expression in the RNAseq data (Supplementary Fig. S1a). Principle component analysis (PCA) of the entire transcriptomes from these RNAseq data showed a clear separation of the three subpopulations (Fig. 1b). In the case of the *MIXL1*-GFP (+)/SSEA-3(+) and *MIXL1*-GFP(−)/SSEA-3(+) cells, data from two biological replicates of each clustered closely, whereas two replicates of the *MIXL1*-GFP(+)/SSEA-3(−) cells were more separated, perhaps reflecting a great heterogeneity and variability expected in populations of differentiated cells. On the other hand, when the PCA was carried out with only a subset of pluripotency associated genes (Fig. 1c) the *MIXL1*-GFP(+)/SSEA-3(+) cluster moved closer to the *MIXL1*-GFP(−)/SSEA-3(+) cluster, especially with respect to PC1 which accounts for 89% of variance, whereas when the PCA was carried out with genes associated with mesoderm (Fig. 1d) the *MIXL1*-GFP(+)/SSEA-3(+) cluster moved closer to the *MIXL1*-GFP(+)/SSEA-3(−) cluster of putatively differentiated cells. These analyses suggest that an active pluripotency network might still be in place the *MIXL1*-GFP(+)/SSEA-3(+) cells, consistent with the possibility that they occupy a substate within the stem cell compartment.

To test this we analysed each subpopulation for the expression of a signature set of genes (Supplementary Table S1), which included 3 controls and 45 genes of interest, by single cell qPCR to assess whether the *MIXL1*-GFP(+)/SSEA-3(+) subpopulation includes individual cells that co-express pluripotency and mesoderm associated genes, and so may represent a transitional substate between that of pristine, pluripotent *MIXL1*-GFP (−)/SSEA-3(+) stem cells and that of mesodermally committed, differentiated *MIXL1*-GFP(+)/SSEA-3(−) cells. The 45 gene set included genes typically associated with the undifferentiated state, such as *OCT4*, *NANOG*, and *SOX2*, and genes typically associated with early mesendoderm differentiation, such as *T(Brachyury)*, *EOMES*, and *GATA6*. Using Monocle2[18] we produced t-distributed stochastic neighbour embedding (t-SNE) analysis of these single cell transcriptomic data. This showed that the *MIXL1*-GFP(+)/SSEA-3(+) cells cluster separately, spanning the space between separate clusters of the *MIXL1*-GFP(−)/SSEA-3 (+) and *MIXL1*-GFP(+)/SSEA-3(−) cells, though there was substantial heterogeneity in each cluster (Fig. 1e).

Single cell analysis did reveal a minor disparity in *MIXL1* expression as reported by GFP positivity and *MIXL1* transcript levels, as assessed by qPCR, which was not evident from the RNAseq data. For example, 7 of 72 cells in the *MIXL1*-GFP(−) fraction exhibited some level of *MIXL1* expression, and conversely a similar level of discordance was seen in the *MIXL1*-GFP(+) cells (Supplementary Fig. S1b). This disparity has been noted previously[11] likely caused by the over 20 h half-life of the GFP[19]. Nevertheless, pseudotime analysis of the single cell qPCR data matched the differentiation trajectory

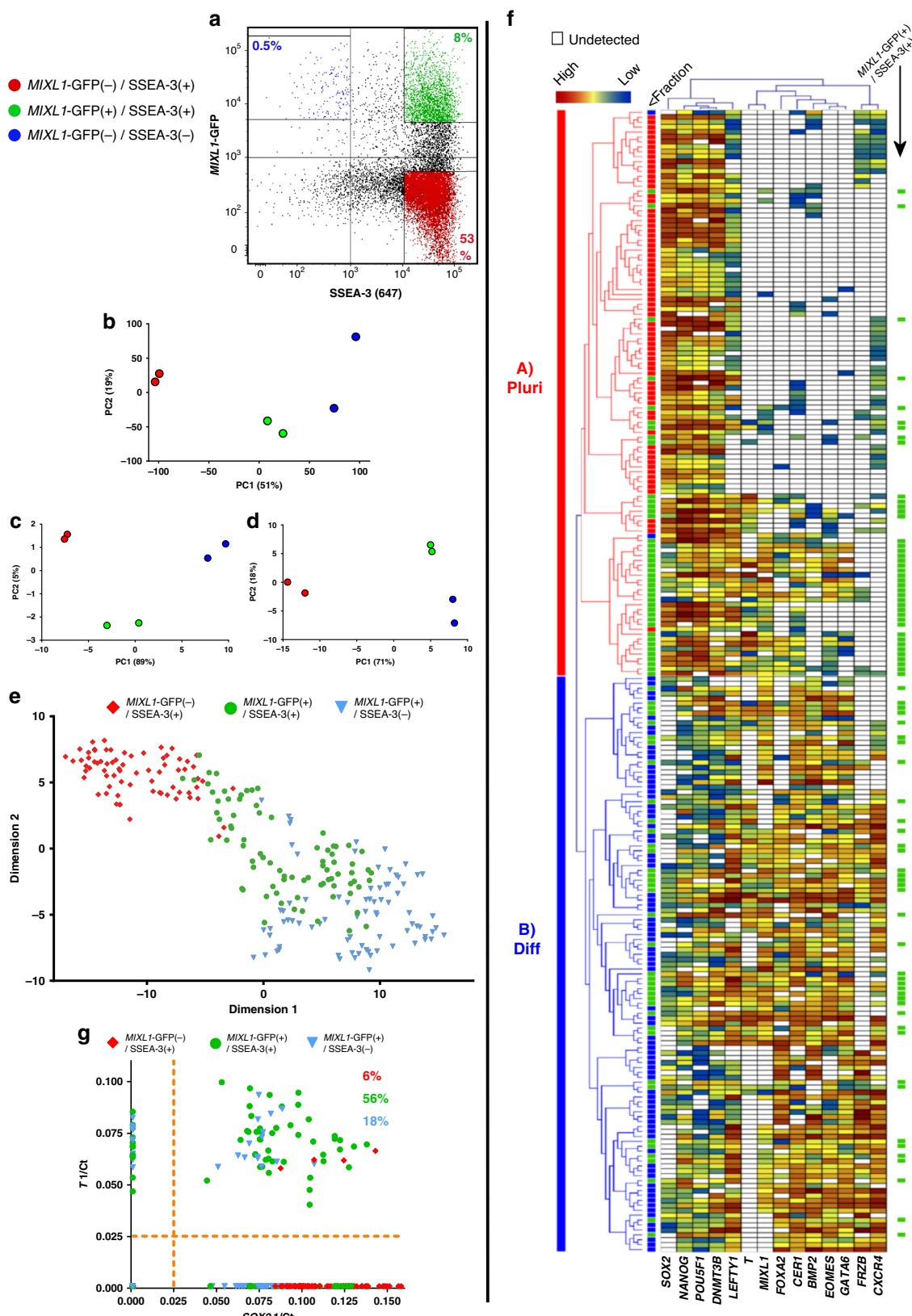

that we inferred from reporter gene and surface marker expression, with *MIXL1*-GFP(−)/SSEA-3(+) mapping to the start of pseudotime and *MIXL1*-GFP(+)/SSEA-3(−) towards the end (Supplementary Fig. S1c, d). Even though the *MIXL1*-GFP and *MIXL1* transcript level is not completely correlated, the use of GFP expression coupled with surface marker expression allowed

for the segregation of states, which pertain to a differentiation trajectory.

A more detailed analysis using a subset of the 48 genes that relate most closely to the pluripotent and differentiated states revealed two main clusters of cells, with the *MIXL1*-GFP (−)/SSEA-3(+) and *MIXL1*-GFP(+)/SSEA-3(−) displaying an

**Fig. 1 A transitionary substate between pluripotency and early mesendodermal differentiation is marked by *MIXL1*-GFP and SSEA-3 coexpression.**
**a** Flow cytometry scatter plot of *MIXL1*-GFP and SSEA-3 expression by HES3-*MIXL1* cells cultured in MEF/KOSR conditions. Coloured populations indicate the fractions isolated by FACS for bulk and single cell transcriptomic analysis, *MIXL1*-GFP(−)/SSEA-3(+) (Red) *MIXL1*-GFP(+)/SSEA-3(+) (Green), *MIXL1*-GFP(+)/SSEA-3(−) (Blue). Principal component analysis (PCA) plots generated from RNAseq data of the *MIXL1*-GFP(−)/SSEA-3(+), *MIXL1*-GFP(+)/SSEA-3(+) and *MIXL1*-GFP(+)/SSEA-3(−)fractions: **b** plot generated from all mRNA expression; plots generated from genes associated with **c** pluripotency and **d** mesoderm. **e** t-SNE plot of the expression of 45 genes (Supplementary Table 1) in individual cells measured by Fluidigm Biomark qPCR. All fractions are displayed in the same dimensional space: *MIXL1*-GFP(−)/SSEA-3(+) cells as red squares, *MIXL1*-GFP(+)/SSEA-3(+) cells as green circles and *MIXL1*-GFP(−)/SSEA-3(+) cells as blue triangles. **f** Heatmap analysis of 232 single cells analysed across 14 of 45 analysed genes using a Fluidigm BioMark system: hierarchical clustering was performed for the genes assessed. The heatmap visualises the individual gene expression after normalisation across genes. White coloured genes indicate undetected levels of expression. Single cells are labelled by colour with the fraction from which they were sorted. Cluster analysis split cells into two broad classification, Cluster A (Pluripotency, Pluri) and Cluster B (Differentiation, Diff). Cells co-expressing pluripotency and differentiation associated genes were readily detected in the *MIXL1*-GFP(+)/SSEA-3(+) fraction but not the other fractions. **g** Scatter plot of 1/Ct values for each single cell for *SOX2* and *T* expression (*MIXL1*-GFP(−)/SSEA-3(+) cells as red squares, *MIXL1*-GFP(+)/SSEA-3(+) cells as green circles and *MIXL1*-GFP(−)/SSEA-3(+) cells as blue triangles). A large proportion of *MIXL1*-GFP(+)/SSEA-3(+) expressed both *SOX2* and *T*. **e**–**g** Source data are provided as a Source Data file.

almost complete separation into Cluster A (Pluripotent) and Cluster B (Differentiated), respectively (Fig. 1f). By contrast, the *MIXL1*-GFP(+)/SSEA-3(+) subpopulation included cells in both of these clusters, with many cells expressing both pluripotency and differentiation associated genes (Fig. 1f). For example, many of the cells in the *MIXL1*-GFP(+)/SSEA-3(+) fraction co-expressed the pluripotency associated marker *SOX2* and the mesoderm associated marker *T* (Fig. 1g). Similar cells were not often seen in the *MIXL1*-GFP(−)/SSEA-3(+) or *MIXL1*-GFP(+)/SSEA-3(−) fractions. The analysis also highlights the clustering of expression of the pluripotency associated genes, *POU5F1*, *SOX2*, *NANOG*, and *DNMT3B*, whereas *MIXL1* clustered most closely with differentiation markers such as *T*, *FOXA2*, and *CER1*. The expression of the full panel of genes assessed is shown in Supplementary Fig. S2. These data suggest that the *MIXL1*-GFP(+)/SSEA-3(+) cells do occupy a transitional substate in terms of gene expression, but do not identify whether that substate is part of the stem cell or differentiated cell compartment.

**MIXL1-GFP(+)/SSEA-3(+) cells contain self-renewing ES cells.** To assess whether the *MIXL1*-GFP(+)/SSEA-3(+) subpopulation contains functional undifferentiated stem cells, we isolated 288 *MIXL1*-GFP(+)/SSEA-3(+) individual cells by FACS and single cell deposition. From these we were able to grow out 47 colonies. Indexed FACS data confirmed the *MIXL1* and SSEA-3 expression levels of the individual cells from which these colonies were obtained (Fig. 2a). All of these 47 clonal colonies exhibited a typical rounded human ES cell colony morphology containing cells with a high nuclear to cytoplasmic ratio, a representative colony is shown in Fig. 2b. From these, 44 colonies were passaged further, and 27 survived; all were positive for the human PSC cell surface antigen TRA-1-81 (Supplementary Fig. S3a). We then randomly selected six clones for further expansion to generate six clonal lines, their original indexed position is shown in Fig. 2a. All displayed high levels of expression for a panel of typical human PSC cell surface antigens, BF4, CD9, SSEA-4, TRA-1-60, TRA-1-81, and TRA-2-49 compared to the negative control P3X (Fig. 2c, Supplementary Fig. S3c). Expression of the pluripotency associated marker, NANOG, was confirmed by intracellular staining (Supplementary Fig. S3b). Although these clonal lines were all derived from *MIXL1*-GFP(+)/SSEA-3(+) cells (Fig. 2a), their pattern of *MIXL1* and SSEA-3 expression in each case had reverted to the pattern seen in the parental cultures rather than retaining the *MIXL1* positive status of the starting cell (Fig. 2d). Thus the *MIXL1*-GFP(+)/SSEA-3(+) subpopulation contains undifferentiated stem cells, despite their transient expression of genes associated with differentiation as cells were able to

repopulate a heterogeneous population (Fig. 2e), and that the *MIXL1*-GFP(+)/SSEA-3(+) and *MIXL1*-GFP(−)/SSEA-3(+) subpopulations represent interconvertible substates of the stem cell compartment.

**Generating MIXL1-GFP(+)/SSEA-3(+) cells in a defined system.** The MEF/KOSR system is not fully defined depending on proprietary components and the use of mouse fibroblasts as a feeder layer and the presence of *MIXL1*-GFP(+)/SSEA-3(+) is inherently variable in this system (Supplementary Fig. S4). We tested whether the *MIXL1*-GFP(+)/SSEA-3(+) subpopulation existed in completely defined conditions, E8 medium[20], with a vitronectin attachment factor culture conditions. We found that this subpopulation vanished and no *MIXL1*-GFP(+)/SSEA-3(+) cells were detected (Fig. 3a). Since we had noted that an inhibitor of endogenous WNT secretion, IWP-2, also inhibited the appearance of *MIXL1*-GFP(+)/SSEA-3(+) cells in MEF/KOSR conditions (Supplementary Fig. S5) similar to findings shown before[21], we inferred that the substate might be regulated by WNT signalling. Therefore, we tested whether activation of WNT signalling in cells cultured in E8 medium could generate this subpopulation. Indeed, the GSK3β inhibitor, CHIR99021 (CHIRON), a canonical WNT signalling mimic, did strongly induce *MIXL1* expression but, although initially the *MIXL1*-GFP(+) cells also expressed SSEA-3, within three days (72 h) many of the cells were *MIXL1*-GFP(+)/SSEA-3(−) (Fig. 3b). However, by the further addition of varying levels of lysophophatidic acid (LPA), a promoter of pluripotency[22,23], we were able to counteract the inductive action of CHIRON and establish conditions in which the pro- and anti-differentiation effects were balanced, so that a substantial proportion of the cells retained expression of both *MIXL1* and SSEA-3 (Fig. 3c). In these exploratory experiments we had also included ROCK inhibitor (Y-27632); subsequently, we found this is unnecessary and, in the absence of the ROCK inhibitor, we established our initial optimal conditions of 3 μM CHIRON and 0.48 μM LPA to form the *MIXL1*-GFP(+)/SSEA-3(+) subset.

Although the combination of CHIRON and LPA was able to generate *MIXL1*-GFP(+)/SSEA-3(+) cells, its ability to maintain this level after passaging was variable. We considered that a possible cause of this variability is the secretion of WNT ligands by cells undergoing early differentiation and that these compromised the level of LPA that we had optimised to counteract the effect of the CHIRON. To address this, we tested a system, similar to the Baseline Activation method proposed in Hackland et al.[24], in which we added the inhibitor of WNT secretion, IWP-2 (Fig. 3d). When the cells were first grown in the presence of 0.48 μM LPA and 3 μM CHIRON and then passaged in the absence of

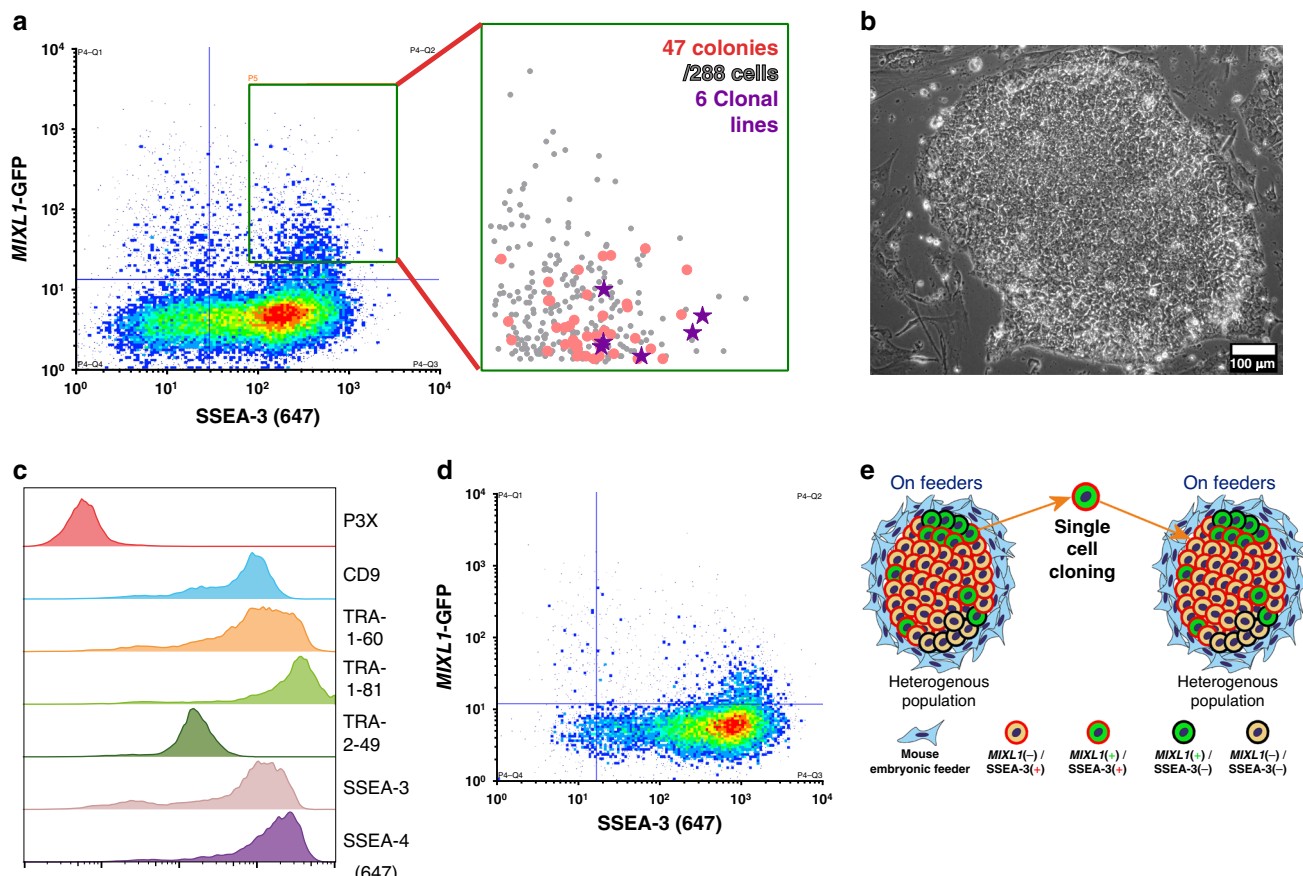

**Fig. 2 The *MIXL1*-GFP/SSEA-3 positive state contains a self-renewing stem cell. a** Density dot plot of *MIXL1*-GFP expression against the surface marker SSEA-3 expression from cells growing in MEF/KOSR conditions. The green gate indicated was used to specify the *MIXL1*/SSEA-3 double positive fraction and cells from this region were sorted into three 96 well plates. The zoomed in scatter plot shows the indexed positions of the 47 isolated cells that formed colonies (red circles), including 6 that were selected randomly to generate clonal lines (purple stars) from 288 sorted cells (grey circles), a cloning efficiency of ~16%. Source data are provided as a Source Data file. **b** Phase contrast image of a typical HES3 *MIXL1*-GFP clone (Clone 2-D2, one of 47 colonies identified) growing in MEF/KOSR conditions (scale bar = 100 μM). **c** Flow cytometry histograms of one of six clonal lines, clone 2-D2, for the stem cell associated antigens CD9, SSEA-3, SSEA4, TRA-1-60s, TRA-1-81 and TRA-2-49 and negative control P3X. All lines displayed similar high expression of these surface markers. **d** Representative density dot plot of *MIXL1*-GFP expression against the surface marker SSEA-3 expression from clonal line HES3 *MIXL1*-GFP 3-C6. All clonal lines re-establish similar *MIXL1*-GFP/SSEA-3 distributions as the starting populations. **e** Schematic diagram of the process of single cell cloning from *MIXL1*/SSEA-3 double positive cells. FACS separates double positive single cells from the heterogeneous population that exists on feeders. Established clonal lines grown on feeders repopulate the heterogeneous distribution seen in the starting population.

IWP-2, there was a marked decrease in expression of SSEA-3. However, when the medium also contained 1 μM IWP-2 there was a striking increase in the expression of SSEA-3 after passaging. These cells from cultures in the presence of IWP-2 also retained expression of other stem cell related antigens and displayed good colony morphology and growth rate over four days post passage. The optimisation process (Fig. 3e) lead to our first medium formulation, which we named PRIMO. The medium consisted of E8 basal medium plus 0.1% BSA, 2 μM cholesterol, 3 μM CHIRON, 1 μM IWP-2 and 0.48 μM LPA. PRIMO medium induced a high proportion of *MIXL1*-GFP (+)/SSEA-3(+) cells in cultures (Fig. 3f).

**Cells in PRIMO medium express early mesendodermal genes.** To assess how closely the 'trapped' *MIXL1*-GFP(+)/SSEA-3(+) subset from cells growing in PRIMO medium resembled the similar subset from MEF/KOSR cultures, we compared the transcriptomes of *MIXL1*-GFP(−)/SSEA-3(+) and *MIXL1*-GFP(+)/SSEA-3(+) cells grown in PRIMO conditions for 3 days, as well as *MIXL1*-GFP(−)/SSEA-3(+) cells from cultures in E8, with the previous data from cells growing in MEF/KOSR conditions.

By PCA, the bulk RNA sequencing data showed a similarity between the *MIXL1*-GFP(+)/SSEA-3(+) cells from both PRIMO and MEF/KOSR cultures whereas the *MIXL1*-GFP(−)/SSEA-3 (+) PRIMO fraction showed some separation from its counter parts in MEF/KOSR and was further separated from the corresponding fraction from E8 cultures (Fig. 4a). We also carried out a single cell qPCR analysis of expression of the 48 gene-set analysed previously. When these data were incorporated into the t-SNE plot previously obtained from the cells growing in MEF/KOSR conditions, the PRIMO-derived *MIXL1*-GFP(+)/SSEA-3 (+) cells showed a distribution that largely superimposed on the *MIXL1*-GFP(+)/SSEA-3(+) cells from MEF/KOSR condition, lying between the *MIXL1*-GFP(−)/SSEA-3(+) and *MIXL1*-GFP (+)/SSEA-3(−) cluster (Fig. 4b). The gene signatures from single cell gene expression patterns from the *MIXL1*-GFP(+)/SSEA-3 (+) fractions from PRIMO and MEF/KOSR cultures were very similar (Supplementary Figs. S6 and S7)

To assess how the *MIXL1*-GFP(+)/SSEA-3(+) cells relate to the trajectory of ES cell differentiation we added 3 μM CHIRON to cells growing in E8 medium and isolated the emerging cell populations at 0, 6, 12, 18, 24, 48 and 72 h, based on *MIXL1*-GFP

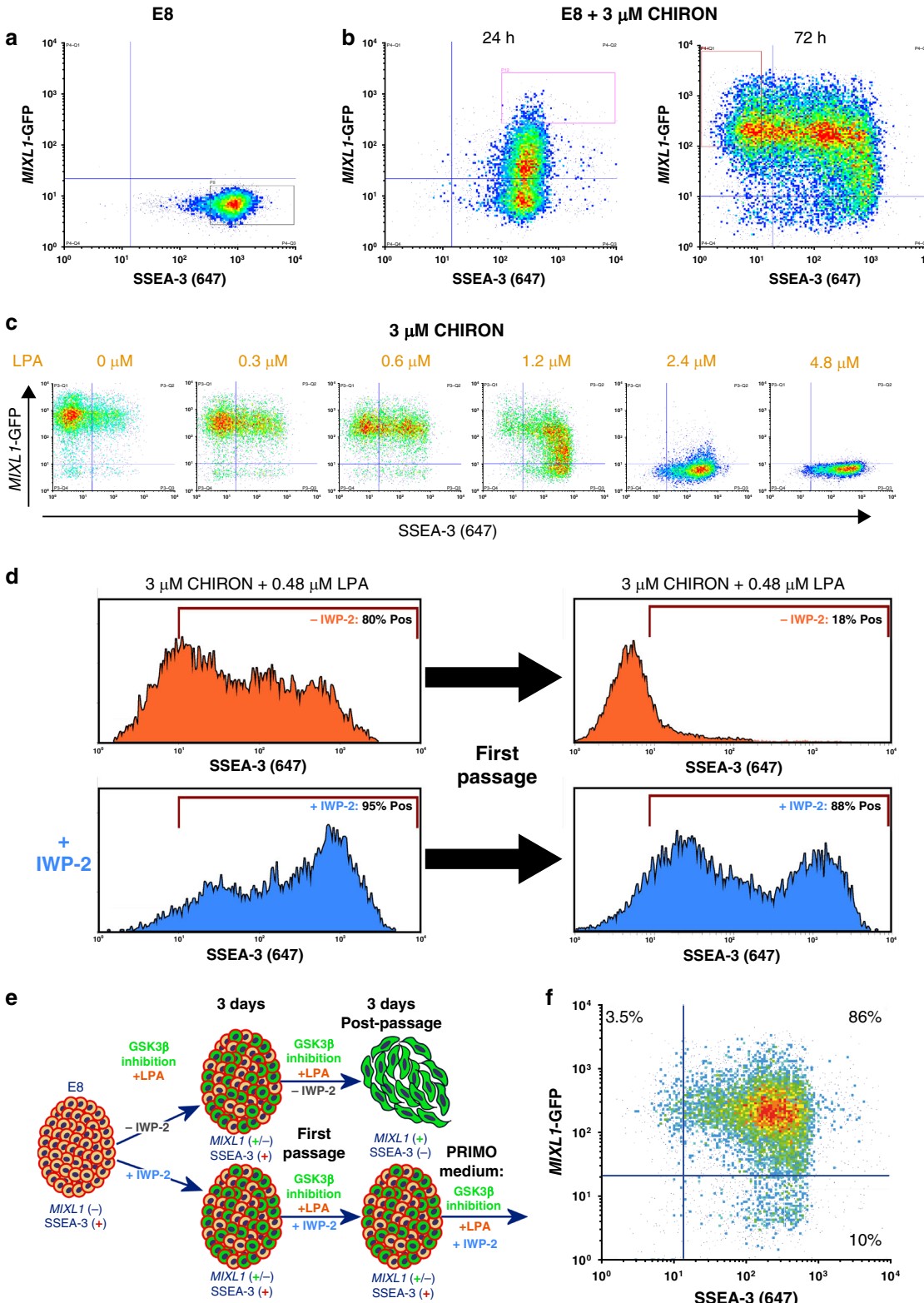

and SSEA-3 expression (Fig. 4c, Supplementary Fig. S8). Over this period the expression of *MIXL1* gradually increased, preceding the eventual loss of SSEA-3 expression. Bulk RNAseq data from the emerging cells at each time point was compared by PCA with RNAseq data from *MIXL1*-GFP(−)/SSEA-3(+) and *MIXL1*-GFP(+)/SSEA-3(+) cells growing in PRIMO conditions for 3 days (Fig. 4d). The *MIXL1*-GFP(+)/SSEA-3(+) fraction mapped between 24 and 48 h, whereas the *MIXL1*-GFP(−)/SSEA-3(+)

fraction from PRIMO conditions did not correlate with the zero hour, *MIXL1*-GFP(−)/SSEA-3(+) cells from E8 but rather mapped on the trajectory between 12 and 18 h of differentiation (Fig. 4d). Thus, the PRIMO medium appears to trap all cells in an intermediate substate but, interestingly, this substate appears to be independent of *MIXL1*-GFP reporter expression. Differential gene expression analysis by DESeq2 and SeqMonk of cells growing in PRIMO, pooling data from both *MIXL1*-GFP(+) and

**Fig. 3 Balancing pro-self-renewal and pro-differentiation signals enables propagation of *MIXL1*-GFP/SSEA-3 cells.** Flow cytometry density plots of SSEA-3 versus *MIXL1*-GFP for HES3 *MIXL1*-GFP cells growing in E8 (**a**) and E8 with 3 μM CHIRON for 24 hours and 72 h (**b**). In E8 SSEA-3 is highly expressed but *MIXL1*-GFP is not detected. The addition of CHIRON drives *MIXL1* expression and eventual loss of SSEA-3. **c** Flow cytometry density plots displaying SSEA-3 expression versus *MIXL1*-GFP expression for HES3 *MIXL1*-GFP cells grown in E8 medium with 3 μM CHIRON and increasing levels of LPA, higher levels of LPA prevent *MIXL1*-GFP expression. **d** Flow cytometry histograms of SSEA-3 expression for cells grown in 3 μM CHIRON + 0.48 μM LPA with IWP-2 (blue) or without IWP-2 (orange). Histograms are from 3 days in culture and 3 days post first passage. SSEA-3 was maintained post passage in samples cultured with IWP-2 present only. **e** A schematic diagram of the development of PRIMO medium. Cells grown under GSK3B inhibition supplemented with LPA and with or without IWP2 induce expression of MIXL1-GFP(+)/SSEA-3(+) cells after 3 days. Post passage without IWP-2 addition cells differentiate, losing SSEA-3, but with IWP-2 cultures maintained SSEA-3 and some MIXL1 expression. **f** After optimisation, the new formulation consisted of 3 μM CHIRON, 1 μM IWP-2 and 0.48 μM LPA. Density plots display the *MIXL1*/SSEA-3 expression under PRIMO revealing high double expression.

*MIXL1*-GFP(−) cells, compared to E8 revealed upregulation of differentiation markers while key pluripotency associated markers such as *NANOG*, *POU5F1* were not differentially expressed (Fig. 4e). Much like the *MIXL1*-GFP(+)/SSEA-3(+) cells from MEF/KOSR (Fig. 1b), our PRIMO cultures showed co-expression of pluripotency associated genes such as *POUF51*, *NANOG*, etc. and differentiation-associated genes such as *MIXL1*, *T* (*BRACHYURY*), *DKK1*, etc. (Fig. 4e). Gene ontology enrichment analysis of the most differentially expressed genes, red squares in Fig. 4e, revealed an enrichment in biological processes such as mesoderm development and gastrulation (Fig. 4f).

**Assessing the lineage bias of cells grown in PRIMO.** After this initial transcriptomic profiling using PRIMO medium, we found that the formulation required further optimisation to increase the maintenance of cells after multiple passages. We developed a further optimised medium containing a higher level of LPA, termed PRIMO Plus which consisted of E8 basal medium plus 0.2% BSA, 4 μM cholesterol, 3 μM CHIRON, 1 μM IWP-2 and 0.96 μM LPA. From our previous work on *MIXL1*-GFP (+)/SSEA-3(+) cells from feeders we predicted that the same population in PRIMO Plus could again generate clonal stem cell lines but also, as our transcriptomic analysis indicated these cells were on a differentiation trajectory, that their differentiation would be biased towards mesoderm. To confirm this HES3-*MIXL1* cells grown for 3 days in PRIMO Plus medium were analysed by cloning and by embryoid body (EB) formation under 'neutral' conditions[25]. The clonal lines established also were used for EB formation. All EBs were then assessed by transcriptome signatures and the score card method[26] (Fig. 5a).

We obtained 38 stem cell-like colonies from 384 *MIXL1*-GFP (+)/SSEA-3(+) cells isolated by FACS and single cell deposition (Fig. 5b). Of these, 31 colonies survived further passaging and were positive for TRA-1-81 staining (Supplementary Fig. S9a). Six of these colonies were randomly selected, initially expanded in MEF/KOSR conditions (Supplementary Fig. S9b, c), transitioned into E8 conditions and then were assessed for their expression of pluripotency associated surface markers expression and *MIXL1*-GFP (Supplementary S9d, e); the index position of the parent cells for each of these expanded lines confirmed that they were derived from *MIXL1*-GFP(+)/SSEA-3(+) cells (purple stars in Fig. 5b). Each of the clones exhibited patterns of antigen expression, and lack of *MIXL1*-GFP, expression, similar to the parental cells growing in E8 conditions (Fig. 5c). These cloning results indicate that the *MIXL1*-GFP(+)/SSEA-3(+) cells from cultures in PRIMO medium do reside within the stem cell compartment and retain the ability to revert to other stem cell substates.

When these PRIMO-derived clones adapted to growth in E8 were cultured as EBs under neutral conditions they showed marked down-regulation of stem cell related genes and upregulation of genes associated with all three germ layers (Fig. 5d), indicating a capacity for multilineage differentiation

with no obvious bias. The transcriptional time course analysis of cells in PRIMO showed that both subsets, irrespective of *MIXL*-GFP expression, are located on the trajectory of mesoderm differentiation induced by CHIRON (Fig. 4d). Therefore, we tested whether these transcriptional changes would lead to differentiation that favours mesoderm formation. When we collected *MIXL1*-GFP(−)/SSEA-3(+) or *MIXL1*-GFP(+)/SSEA-3(+) cells by FACS or unsorted cells grown in PRIMO Plus medium, and grew them as EBs under neutral conditions, they all showed a marked bias to mesoderm differentiation, irrespective of *MIXL1* expression. Congruent with the transcriptional time course analysis, EBs showed that cells grown in PRIMO medium are on a mesoderm trajectory which impacts upon their lineage fate decision, an impact that was reversible when clonal lines were established in E8 conditions (Fig. 5d).

Since the cells grown in PRIMO Plus exhibited a mesoderm bias we also assessed their ability to generate haematopoietic progenitors, as an example of a particular lineage derived, in this case, from lateral plate mesoderm. In addition to HES3-*MIXL1* we used another human PSC line, H9 *T*-Venus, that carries a Venus reporter for *T* (*BRACHYURY*) expression[27] for this assay. Cells were first passaged into flasks containing either E8 or PRIMO Plus media. After three days the cells were replated onto confluent OP9 mouse stromal cells, while high co-expression of reporter gene with SSEA-3 and TRA-1-81 was confirmed by analysis of a sister flask (Supplementary Fig. S10). When analysed ten days later, for both cell lines, 5–15% more cells from the PRIMO cultures than from E8 cultures expressed haematopoiesis-associated molecules CD43 and CD34, assessed by flow cytometry (Fig. 6a). Taking into account the final numbers of harvested cells, this equated to a 30–40% increase in yield (Fig. 6b) of CD43+ cells generated in PRIMO compared to the E8 conditions (Fig. 6c).

We next used directed EB differentiation towards alternative germ layers[28] to test whether the cells grown in PRIMO Plus retain full pluripotency, despite their bias towards mesoderm. Using conditions to direct the EB differentiation of HES3-*MIXL1* or H9 *T*-Venus cells grown in PRIMO media for three passages towards ectoderm, mesoderm or endoderm we observed respective lineage signatures, assessed by the hPSC scorecard method[26] (Fig. 6d). To test further the potency of cells maintained in PRIMO we also assessed whether they could be induced using a very different, 2D culture method[24] to differentiate into neural crest, a derivative of the embryonic ectoderm. We were able to generate SOX10 positive neural crest cells from all three cell lines grown in PRIMO medium (Fig. 6e and Supplementary Fig. S11) at efficiencies within the published range[24,29]. We also found that the use of the WNT inhibitor, IWP-2, during differentiations towards ectoderm and endoderm aided efficient differentiation (Fig. 6d), as previously shown[13,30]. This was also the case for neural crest formation (Supplementary Fig. S11c). Together, these results confirm that cells maintained in

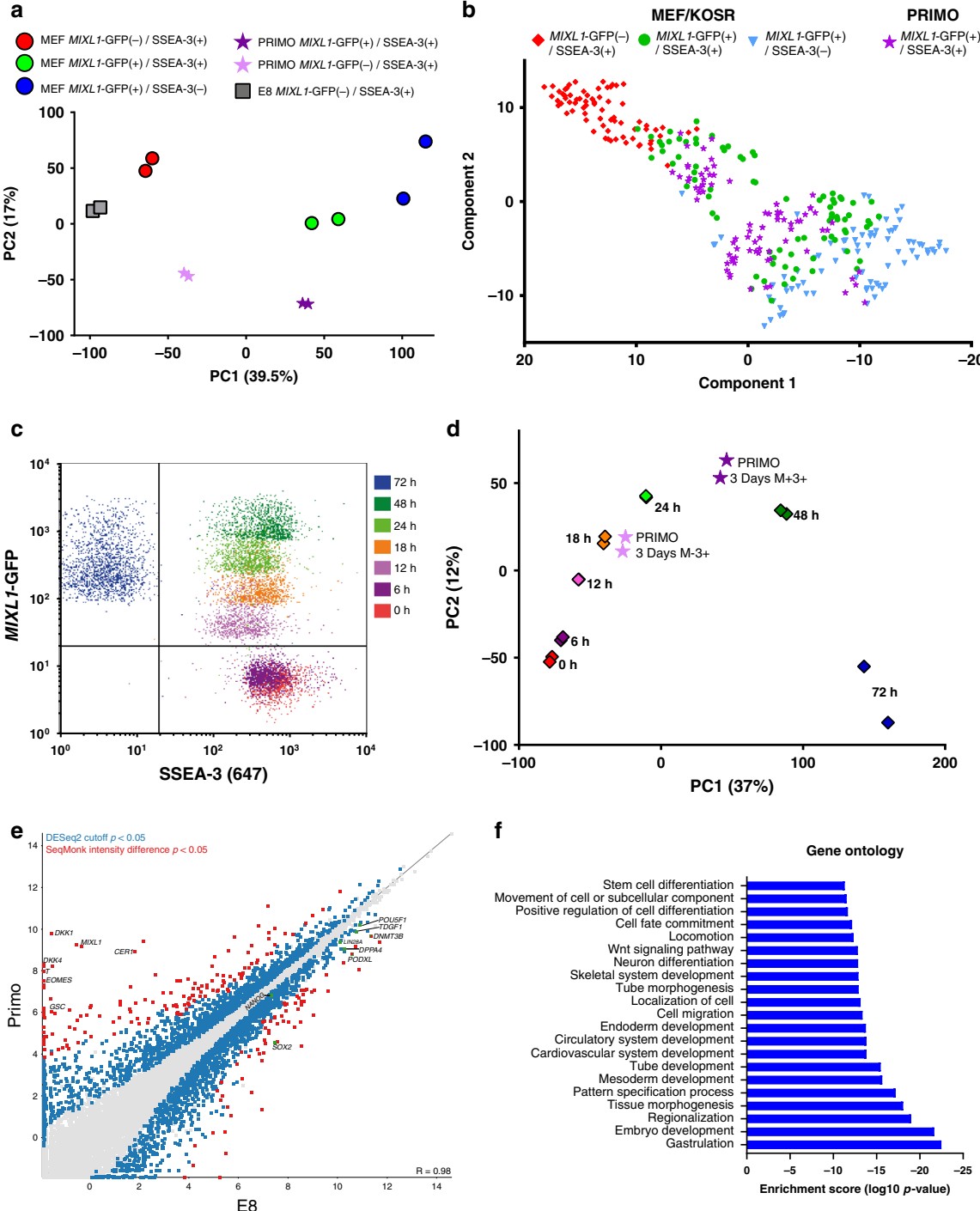

**Fig. 4 Cells in PRIMO transcriptionally correlate with the MEF/KOSR substate and early mesodermal differentiation. a** Principal component analysis of *MIXL1*-GFP(+)/SSEA-3(+) (dark purple stars) and *MIXL1*-GFP(−)/SSEA-3(+) (light purple stars) cells grown in PRIMO compared to the fractions identified in MEF/KOSR conditions (circles) and standard E8 culture (grey squares). **b** t-Distributed Stochastic Neighbour Embedding (t-SNE) plots of the expression of 45 genes in individual cells measured by Fluidigm Biomark qPCR and analysed by Monocle2. Displays all fractions in the same dimensional space from MEF/KOSR conditions (presented in Fig. 1c) *MIXL1*-GFP(−)/SSEA-3(+) cells as red squares, *MIXL1*-GFP(+)/SSEA-3(+) cells as green circles, *MIXL1*-GFP(−)/SSEA-3(+) cells as blue triangles and PRIMO conditions *MIXL1*-GFP(+)/SSEA-3(+) cells as purple stars. **c** Flow cytometry dot plot of the emerging population for which was sorted at the indicated timepoints during a differentiation time-course. Differentiation was induced incubating cells with E8 containing CHIRON at 3 μM for 72 h. **d** Principal component analysis of *MIXL1*-GFP(+)/SSEA-3(+) (dark purple stars) and *MIXL1*-GFP(−)/SSEA-3(+) (light purple stars) cells grown in PRIMO compared to the differentiation time course (coloured squares). **e** Gene expression scatter plot of bulk RNAseq analysis of cells grown in E8 compared to PRIMO medium. Differentially expressed genes were identified by DESeq2 with a cut off *p* value of <0.05 (blue squares) and the SeqMonk intensity difference filter with a *p* value cut off <0.05 (red squares). Some key genes related to pluripotency are shown as green squares. **f** Gene ontology (GO) statistical enrichment analysis was performed by ToppGene[64] and Revigo[62] on the list of genes from the SeqMonk[59] intensity difference filter. GO terms which were significantly enriched are shown, the Log10 *p* values for each GO term are displayed. GO terms related to early differentiation and gastrulation are significantly enriched. **b**, **e** Source data are provided as a Source Data file.

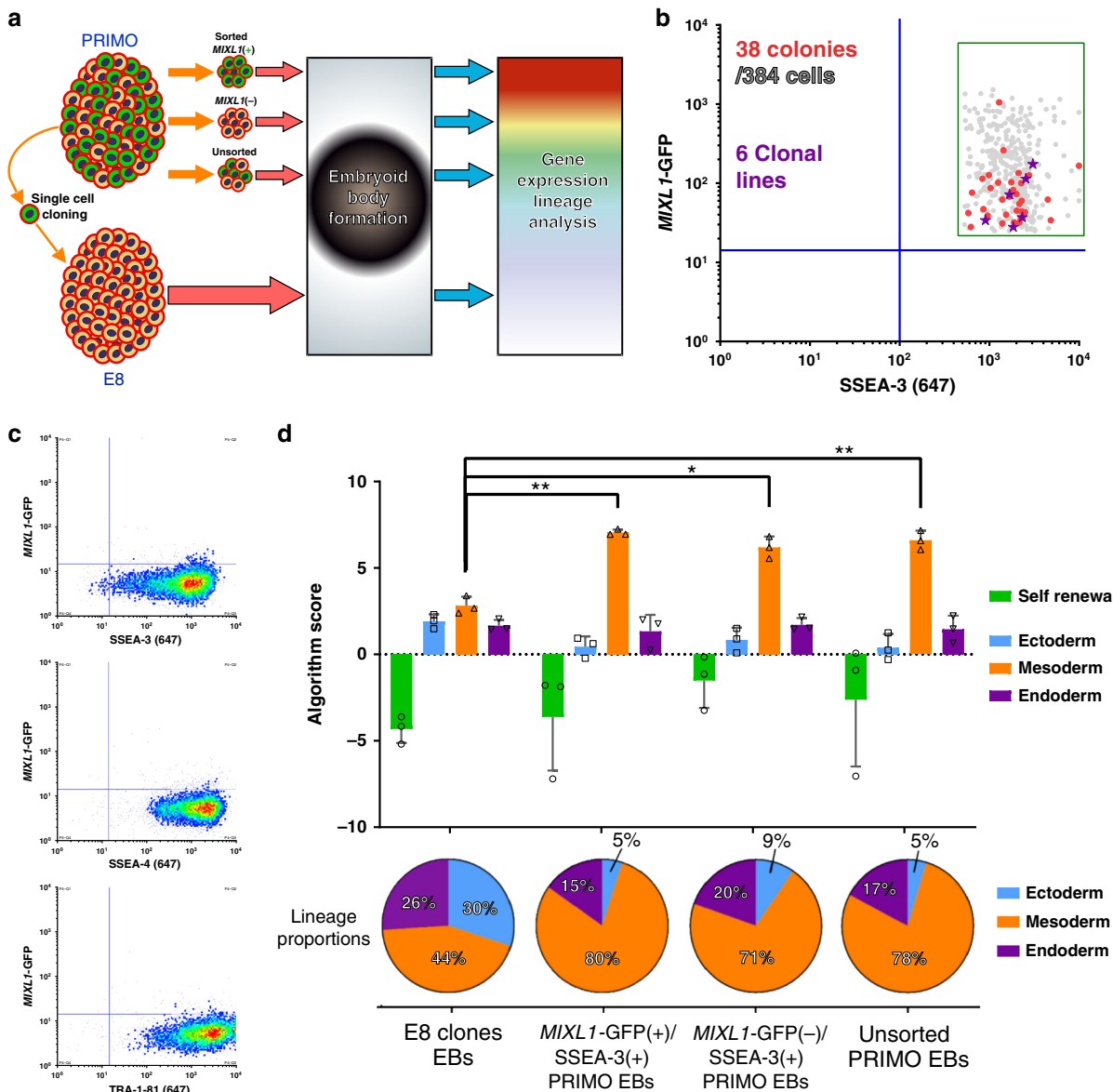

**Fig. 5 Cells in PRIMO exhibit a mesoderm bias under differentiation conditions and retain clonogenic potential. a** A schematic diagram of the experimental process. Cells were grown in PRIMO Plus conditions for 3 days then cells were taken, unsorted or sorted for *MIXL1*-GFP(+)/SSEA-3(+) and *MIXL1*-GFP(−)/SSEA-3(+), and put through a neutral EB assay. After 7 days EBs were harvested and assessed for gene expression by hPSC scorecards. Separately single *MIXL1*-GFP(+)/SSEA-3(+) cells were sorted from cells grown in PRIMO conditions and clonal lines were established, these lines were then assessed by the same neutral EB assay. **b** Single cells from *MIXL1*/SSEA-3 double positive fraction were sorted into four 96 well plates. The *MIXL1*/SSEA-3 scatter plot shows the indexed positions of the 38 isolated cells that formed colonies (red circles), including 6 that were selected randomly to generate clonal lines (purple stars) from 384 sorted cells (grey circles), a cloning efficiency of ~10%. **c** Flow cytometry density plots of clone 10-A4 grown in E8 conditions for surface markers SSEA-3, SSEA-4 and TRA-1-81 versus *MIXL1*-GFP, revealed high surface marker expression and virtually no *MIXL1*-GFP expression. **d** A bar chart displaying the algorithm score for each sample, for self renewal and three lineages, ectoderm, mesoderm and endoderm. The algorithm score is calculated based on the qPCR values for genes of a given lineage, the 0 baseline is based on the average value of undifferentiated samples. EB samples have been normalised to their undifferentiated counterparts (bars are mean ± SD, $n = 3$ biological replicates, significance assessed by two-way ANOVA analysis corrected for multiple comparisons, p values left to right are (\*\*) $p = 0.0026$, (\*) $p = 0.0177$ and (\*\*) $p = 0.0071$). Below the bar chart are lineage proportion pie charts of the algorithm scores generated for the three germ layers. All samples grown in PRIMO medium generated EBs with enhanced mesoderm signatures, whereas clonal lines grown in E8 generated EBs containing all three germ layers with more similar proportions. **b**, **d** Source data are provided as a Source Data file.

PRIMO retain full pluripotency, despite their mesoderm bias under neutral differentiation conditions.

**PRIMO Plus maintains a mesoderm biased pluripotent state.** To confirm that we could maintain human PSC in a biased substate through successive passages, we analysed HES3-*MIXL1* ES cells after three passages in PRIMO Plus medium. In addition,

we also transitioned into PRIMO Plus medium another human PSC line, H9 *T-Venus* and a human IPS cell line, MIFF1[31], and likewise maintained them through three passages. In each case, the cells retained an undifferentiated morphology and expressed high levels of SSEA-3 (Fig. 7a–c). The HES3-*MIXL1* cultures retained, as before, a population of *MIXL1*-GFP(+)/SSEA-3(+) cells (Fig. 7a), while the H9 *T-Venus* cultures contained a

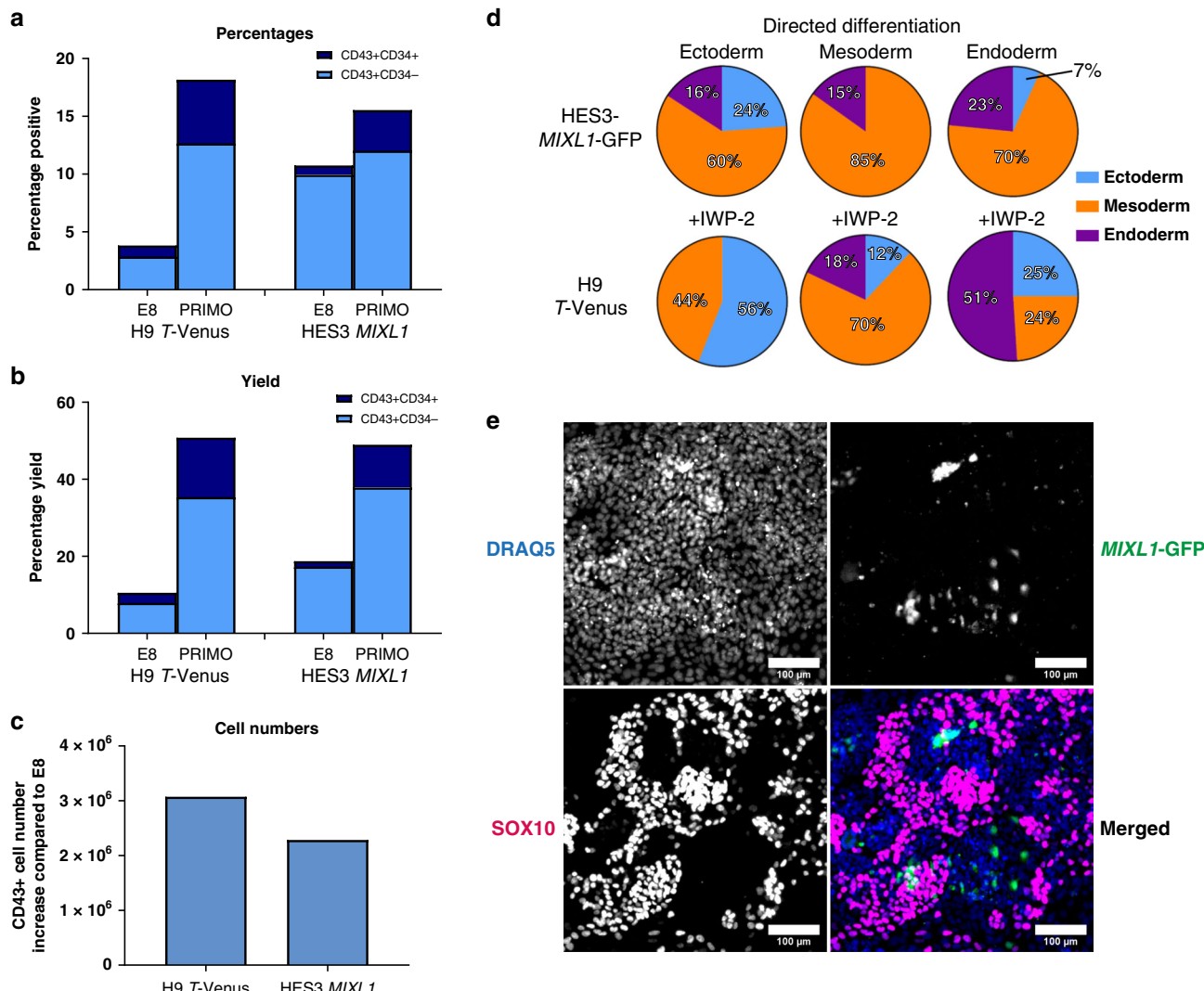

**Fig. 6 Cells in PRIMO exhibit increased differentiation potential towards mesodermal derivatives and the plasticity to generate other lineages.** Cells grown in E8 or PRIMO Plus were subjected to a blood differentiation protocol, on Day 10 CD43 and CD34 expression was quantified by flow cytometry. **a** Bar chart shows the percentage positive cells from each cell line and starting condition. **b** Bar chart shows percentage yield, calculated based on the number of cells from a sister flask at the start of the differentiation, from each cell line and starting condition. **c** Bar chart shows CD43+ cell numbers calculated based on the cell count and percentage positive and compared to the E8 condition for each cell line (a-c bars are values from one biological replicate from two independent cell lines). **d** Lineage proportion pie charts of the algorithm scores generated for the three germ layers under directed differentiation protocols for ectoderm, mesoderm and endoderm. HES3 *MIXL1*-GFP and H9 *T*-Venus were grown in PRIMO PLUS for 3 and 4 passages, respectively prior to EB formation, the WNT inhibitor IWP-2 was also added at 1 μM in H9 *T*-VENUS conditions (pie charts are generated from one biological replicate from two independent cell lines). **a–d** Source data are provided as a Source Data file. **e** Representative immunofluorescent images of SOX10 expression at day 5 of neural crest differentiation of HES3-*MIXL1* from PRIMO Plus cultures. A merged image of all three channels is present DRAQ5 (Blue), *MIXL1*-GFP (Green) and SOX10 (Magenta) (Scale bar = 100 μm, ~100,000 cells were analysed in total from one biological replicate, two further cell lines were analysed in Supplementary Fig. S9).

substantial population of *T* (*Brachyury*) (+) cells, this population being more consistently abundant than the *MIXL1*-GFP(+) population (Fig. 7b), whereas in E8 medium there were no *MIXL1*-GFP(+) (Fig. 3a) or *T* (*Brachyury*) (+) cells (Supplementary Fig. S12a). These cells also expressed a typical set of pluripotency-associated genes that were down regulated when the cells were allowed to form EBs under neutral differentiation conditions. At the same time, the pattern of induced gene expression indicated a strong mesoderm bias in the differentiation of each of the lines (Fig. 7d). These results confirm that in the optimised PRIMO Plus medium, human pluripotent stem cells can be maintained following passaging in a mesoderm biased substate within the stem cell compartment.

PRIMO Plus remained our optimal medium, but we also tested whether we could substitute components for others that targeted the same or similar pathways. Using H9 *T*-Venus reporter we found that recombinant Dickkopf WNT signalling Pathway Inhibitor 1 (DKK1), which inhibits WNT ligand binding to WNT receptors, could be used at 100 ng/mL in place of IWP-2 (Supplementary Fig. S12b). Similarly, CHIR99021 could be substituted with the GSK3β inhibitor SB216763 at 10 μM (Supplementary Fig. S12c). For replacement of LPA we focused on activators of G-Coupled Protein receptors. We used various concentrations of Sphingosine-1-phosphate (S1P), another phospholipid present in KOSR medium that has been shown to inhibit differentiation[23,32], and a chemical agonist for LPA receptor 2

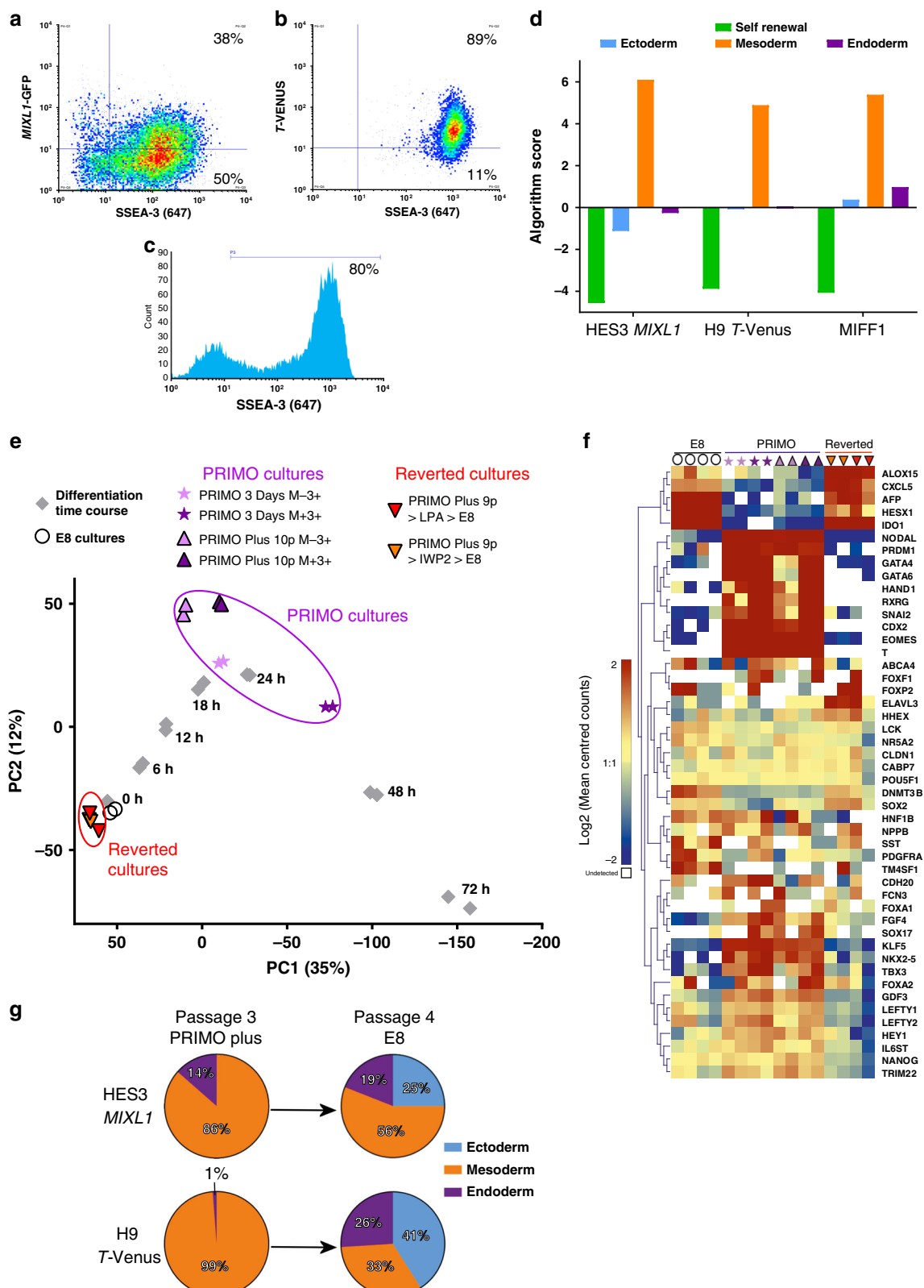

(GRI977143)[33]. The cell expression patterns observed using S1P or GRI977143 media were similar to that observed using LPA-based media, albeit at slightly different concentrations (Supplementary Fig. S12d, e).

We returned to our PRIMO Plus formulation to assess the phenotypic and genetic stability of these lines in extended culture. The results showed that the pluripotency associated surface markers SSEA-3, SSEA4, THY1 and TRA-1-81 were highly expressed in HES3-MIXL1 at passage 7 and H9 T-Venus at passage 10 (Supplementary Fig. S13a, b). H9 T-Venus cells were also assessed at passage 3 in PRIMO Plus for their NANOG and SOX2 expression by immunofluorescence, revealing high co-expression of these markers with T-Venus (Supplementary Fig. S14). Further, both HES3-MIXL1 and H9 T-Venus cells

**Fig. 7 Cells in PRIMO medium can be maintained in a mesoderm biased state and revert back to an unbiased state.** Flow Cytometry analysis of cells grown in PRIMO Plus for three passages. **a** Density plot of *MIXL1*-GFP vs SSEA-3 for HES3 *MIXL1* **b** Density plot of *T*-Venus vs SSEA-3 for H9 *T*-Venus **c**. Histogram of SSEA-3 expression for MIFF1. **d** Algorithm score for EBs generated from HES3-MIXL1, H9 *T*-Venus and MIFF1 (iPS line) grown in PRIMO medium for three passages. All lines exhibited strong mesoderm signatures under neutral conditions (*n* = 1 biological repeat from three separate lines). **e** Principal component analysis of cells grown in PRIMO for 3 days (purple stars) or ten passages (purple triangles) compared to the differentiation time course (grey squares). Also shown are cells, which had been grown in PRIMO for nine passages and then transitioned back into E8 conditions (red and orange triangles). Cells maintained in PRIMO cultures are close to 18–24 h of differentiation and reverted cultures close to 0 h (E8 only) cultures (white circles). **f** Heatmap analysis of key genes related to Pluripotency and mesendoderm differentiation generated with Log2 counts that have been mean centred per gene with a colour scale from −2 to 2. **g** Lineage proportion graphs from neutral EBS generated from HES3-*MIXL1* and H9 *T*-Venus after three passages in PRIMO Plus and subsequent reversion into E8 medium. The strong mesoderm bias was alleviated upon transition back into standard conditions. **b**, **d**, **e** Source data are provided as a Source Data file.

growing in PRIMO Plus maintained a normal diploid karyotype after ten passages, assessed by G-banding and scoring 30 metaphase spreads (Supplementary Fig. S15)

To assess further how the long term passaged, mesoderm biased cells related to the trajectory of mesoderm differentiation induced by CHIRON we performed bulk RNA-seq analysis on the *MIXL1*-GFP(−)/SSEA-3(+) and *MIXL1*-GFP(+)/SSEA-3(+) HES3-*MIXL1* cells from cultures maintained in PRIMO Plus for 10 passages (Supplementary Fig. S13c). We compared these data (Fig. 7e) to our previous analysis of the time course of differentiation induced by CHIRON and also cells cultured PRIMO cultures for just 3 days (Fig. 4d). The transcriptome of both the *MIXL1*-GFP(+)/SSEA-3(+) and *MIXL1*-GFP(−)/SSEA-3(+) cells after ten passages in PRIMO Plus correlated with cells located between 18 and 24 h on the differentiation trajectory, similar to the previously assessed cells after 3 days in PRIMO medium.

Further, we show crucially that when cells were transitioned back into E8 medium after nine passages in PRIMO Plus, these induced transcriptional changes reverted to expression levels typical of cells maintained in E8 medium. At the same time, *MIXL1*-GFP/SSEA-3 expression similarly reverted (Supplementary Fig. S13d). Assessing a sample of genes related to pluripotency and mesendoderm differentiation reveals a pattern of expression shared by PRIMO cultures and lost in reverted cultures. Mesendoderm genes such as *T, EOMES, NODAL*, etc. are all highly expressed in PRIMO cultures and lowly or non-expressed in both E8 and reverted cultures. Pluripotency associated genes such as *NANOG, POU5F1, SOX2* and *DNMT3B* have less varied expression levels across all samples (Fig. 7f), the protein expression of NANOG and SOX2 was also validated by immunofluorescence in PRIMO Plus and reverted cultures (Supplementary Fig. S16). The growth rates and morphology of cells were also monitored by time lapse microscopy showing consistent growth in all conditions. (Supplementary Fig. S17).

Another key question is the extent and ease with which mesoderm biased stem cells that have been maintained for multiple passages in PRIMO medium revert to a non-lineage biased pluripotent state when transferred back to standard E8 medium. When both HES3-*MIXL1* and H9 *T*-Venus cells were transferred to E8 medium after multiple passages in PRIMO, SSEA-3 expression was maintained but reporter expression was substantially reduced highlighting the plasticity of the mesoderm biased stem cell state (Supplementary Fig. S18). To confirm that the transcriptional reversion seen upon the transfer of cells from PRIMO to E8, was accompanied by functional reversion, neutral EBs were generated from PRIMO Plus and reverted cultures. Whereas with both cell lines a strong mesoderm bias was seen in, the EBs produced from cells kept in PRIMO Plus, the bias was diminished in the cells transferred to E8 medium with strong gene expression signatures from all three germ layers (Fig. 7g). This demonstrates how easily cells can move through these

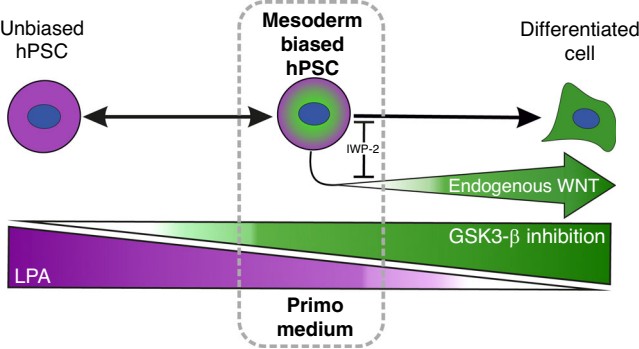

**Fig. 8 A schematic diagram of the trapping of a Mesoderm Biased Human PSC (hPSC).** The cross antagonism of the pro-pluripotency factor (LPA) and pro-differentiation factor (GSK3β Inhibition) is able to induce a mesoderm biased state in human PSC but endogenous WNT secreted by the cells drives further differentiation. Using IWP-2 to block this endogenous WNT secretion prevents this further differentiation. These three components of PRIMO medium permit the maintenance of the mesoderm biased state in vitro.

substates of pluripotency, leading to functional consequences in their differentiation potential.

## Discussion

One view to accommodate the apparent heterogeneity of human PSC is that the stem cell compartment comprises several inter-convertible substates in which the cells retain pluripotency but behave transiently as differentiation intermediates, with distinct propensities for differentiation and lineage selection. We have now identified a mesoderm biased state within human plur-ipotent stem cell compartment, which we can induce and main-tain in PRIMO, a defined medium based on cross antagonism, i.e. simultaneously applying signals that promote differentiation (GSK3-β inhibition) and pluripotency (LPA) (Fig. 8). This state exhibited characteristics corresponding to undifferentiated cells, demonstrated through their capacity to generate clonally derived lines that retain pluripotency. Examining the transcriptome, by bulk and single cell approaches, highlighted co-expression of pluripotency and differentiation associated gene regulatory net-works within the substate. This substate could be induced in three independent human PSC lines and remained stable over multiple passages.

The cells induced by PRIMO medium appear to be relevant normal intermediates in the differentiation of human PSC towards mesoderm, as their transcriptome is positioned between 18 and 24 h on the time course of induced mesoderm differ-entiation. Although initially we defined the state using *MIXL1* expression, cells in PRIMO were heterogeneous for *MIXL1*-GFP expression whereas *T(Brachyury)* was more widely expressed as judged by the *T*-Venus reporter. When compared to our

differentiation time course *MIXL1*-GFP(+) and *MIXL1*-GFP(−) cells both sit on the trajectory with the *MIXL1*-GFP(+) cells sitting slightly further on the trajectory. Importantly *MIXL1*-GFP(−) cells grown in PRIMO, like their *MIXL1*-GFP(+) counterparts, displayed a strong mesoderm bias in neutral EB assays, in contrast to the *MIXL1*(−) cells grown in E8 medium.

LPA and CHIRON have been used in combination before by Blauwkamp et al.[22] in an attempt to enhance differentiation efficiency through reducing heterogeneity of WNT expression in human ES cultures. Our goal was to trap cells in a mesoderm biased intermediate state using LPA and CHIRON but CHIRON induction produces a positive feedback loop, since the derivative cells themselves produce WNT ligands which drives further differentiation. Indeed, our single cell transcriptomic analysis, showed expression of endogenous WNT related genes, consistent with the notion that early differentiation would lead to the secretion of WNT ligands which would confound our ability to precisely regulate WNT activation with CHIRON. To counteract this positive feedback loop, we reasoned that we could incorporate a Baseline Activation approach[24].

Baseline Activation as an approach focuses on blocking the endogenous signalling that allows an exact titration of the signal by addition of exogenous agonist, affording tighter control. We used IWP-2 to inhibit WNT ligand secretion, or DKK1 to block WNT ligand binding, while activating the pathway with CHIRON or SB216763. This strategy enabled us to stabilise cultures in PRIMO medium and maintain the cells in this transitory pluripotent state that exhibited mesoderm bias (Fig. 6e). Bakre et al. described a seemingly similar mesoderm biased state in mouse PSC induced into a mesoderm biased state through manipulation of WNT signalling[34]. However, reversion of their cells to an unbiased state took two weeks of culturing. While the generation or identification of multipotent progenitors towards mesoderm and endoderm from PSC has been demonstrated previously in human[35−37] PSC in vitro, these studies indicated a loss of traditional pluripotent markers such as NANOG, POU5F1 and SOX2 and a lineage restricted differentiation potential, representing trapping of later stages of differentiation. An important distinction is that the substate we describe here is plastic with cells being able to further differentiate or, when returned to E8 medium, readily revert to an unbiased stem cell state demonstrated both by reversion of the transcriptional profile and reacquisition of unbiased differentiation in EB assays. Nevertheless, retention of pluripotency of the cells in the substate was shown by their ability to generate ectoderm, endoderm or neural crest, under appropriate differentiation conditions.

Taken together our results suggest that the intermediate we have trapped sits on a normal in vitro differentiation trajectory the investigation of which may lend further insight into the critical events that ultimately lead to the exit from pluripotency. How this intermediate relates to the differentiation within the human embryo remains unclear. Interestingly, cells in PRIMO express certain lineage markers including *T* and *MIXL1* that are expressed in a subset of mouse epiblast stem cells that are derived from the post-implantation embryo[38−40]. It is plausible that this similarity in the transcriptome reflects correspondence to a similar developmental intermediate in the human embryo. These considerations aside, we wish to emphasise that these cells represent a key in vitro intermediate in the differentiation of human PSC that has practical implications for production of cells for use in regenerative medicine. In considering the application of this medium in translation medicine, we envisage that limited passaging in PRIMO would be sufficient to bias a population of human PSC prior to differentiation. From our experience of growing cells in PRIMO Plus the generation of more than $10^{10}$ cells after even only three passages from an initial seeding of one

million cells is possible. Whilst we have demonstrated cells remained genetically stable over ten passages suggesting the medium may be safe for scale up, the state induced appears similar even after a short period in the medium suggesting extended passaging in PRIMO may not be necessary. The system of cross-antagonism within the controlled environment of baseline activation that we have used to generate a mesoderm-biased substate of human PSC could, in principle, be used to generate other lineage-biased substates or facilitate the capture of relevant differentiation intermediates at later developmental stages.

## Methods

**Cell culture**. Four human PSC lines were used in this project, ES cell lines, HES3[41], HES3-*MIXL1*[11] and H9 *T*-Venus[27], and iPSC line, MIFF1[31]. Both feeder and feeder free systems were used. For either system, flask/plates of hESCs were grown in humidified incubators at 37 °C and 5% CO$_2$. Our feeder culture system uses a layer of mitotically inactivated mouse embryonic feeders and KOSR medium consisting of KnockOut DMEM (ThermoFisher), 20% KnockOut Serum Replacement (Thermofisher), 1x Non-essential amino acids (ThermoFisher), 1mM L-Glutamine (ThermoFisher), 0.1 mM 2-Mercaptoethanol (ThermoFisher) and 4 ng/ml FGF2 (Peprotech) and is referred to as MEF/KOSR conditions. For feeder free systems we used two matrices Geltrex (ThermoFisher) and Vitronectin (Stem cell Technologies). This was combined with E8 medium (made in house, adapted from Chen et al.[20]) with L-Glutamine being replaced for the more stable GlutaMax (ThermoFisher). For LPA containing and subsequent PRIMO media, the LPA media is first made as a 10× stock in E8 medium, 10× BCL (BSA, Cholesterol, LPA) at 1% Fatty acid Free BSA (Probumin, Millipore), 20 μM Cholesterol (Synthechol, Sigma) and 4.8 μM Oleoyl-L-α-lysophosphatidic acid sodium salt (LPA, Sigma). This was then diluted in E8 as needed for PRIMO and PRIMO Plus, then CHIRON 99021 (Tocris) and IWP-2 (Tocris) were added at 3 μM and 1 μM, respectively. A detailed protocol for medium preparation and passaging in PRIMO Plus has been uploaded to Nature protocol exchange[42]. Cells for MEF/KOSR were passaged by treatment with Collagenase IV (ThermoFisher) and manually scrapped away. Cells for feeder free systems were passaged with a non-enzymatic disassociation solution ReLeSR (Stem Cell Technologies) according to manufacturer's instructions.

**Flow cytometry**. Single cell suspensions were harvested by treating with accutase (ThermoFisher) for 10 min, then washed and resuspended in Dulbecco's Modified Eagle's medium (DMEM) and 10% Foetal Calf Serum (FCS) at a density of $1 \times 10^7$ per mL. Antibodies were added at the appropriate dilution. After addition of the primary antibody cells were incubated at 4 °C for 30 min. Cells were then washed with DMEM/FCS and resuspended in DMEM/FCS. Secondary antibody, Goat anti Mouse Affinipure IgG + IgM (H + L) Alexafluor 647 (Jackson Laboratories) at 1:200 and incubated at 4 °C for 30 min. Cells were then washed and resuspended in fresh DMEM/FCS for flow cytometry analysis. To set baselines for *MIXL1*-GFP and negative secondary 647, unlabelled HES3 line was harvested and stained for P3X. P3X is an IgG1 antibody that is secreted from the parent myeloma, P3X63Ag8, from which all in house hybridomas had been derived[43]. P3X shows minimal reactivity to human cells[43]. Positive gates were set according to HES3 P3X negative controls. Samples were also stained for P3X to assess non-specific binding. All flow cytometry analysis contained P3X samples for baseline setting (source data). The monoclonal antibodies BF4[44], CD9(CH8)[44], SSEA-3[45], SSEA-4[46], THY1 (CD90)[47], TRA-1-60[48], TRA-1-81[48] and TRA-2-49[49] were prepared in house from the relevant hybridomas as previously described[16,47].

**Intracellular staining**. Cells were fixed using 4% PFA for 15 min and permeabilised with 0.5% Triton X-100 in PBS (w/o Ca+, Mg+ +) with 10% FCS and 0.1% BSA for 10 min. Cells were washed and then incubated with blocking buffer consisting of 10% FCS and 0.1% BSA in PBS (w/o Ca+, Mg+ +) for one hour. Cells were then washed, primary antibody was added and they were incubated overnight at 4 °C. Antibodies used were: Anti-NANOG (XP® #4903, Cell Signalling Technology, 1:400), ANTI-SOX2 (XP® #3579, Cell Signalling Technology, 1:400) and Anti-SOX10 (Cell Signalling Technology, 89356S, 1:500). Secondary antibodies, Goat anti Mouse Affinipure IgG+IgM (H + L) Alexafluor 647 (Jackson Laboratories) or Goat anti Rabbit Affinipure IgG+IgM (H + L) Alexafluor 594 (Jackson Laboratories) at 1:200 and incubated at 4 °C for 2 h min. Hoescht 33342 (ThermoFisher, #H3570) was added at 1:10000 or DRAQ5 at 1:1000 to the diluted secondary antibody solution. Samples were imaged on an INCell 2200 (GE Healthcare) and analysed on Cell Profiler v2.2[50]. Density plots of expression were generated using FlowJo v10.6[51].

**Live TRA-1-81 staining**. In order to assess the progress of the clonal line formation we performed live staining for a pluripotency-associated surface marker TRA-1-81. Lines were assessed after the first passage into 48 well plates. TRA-1-81 antibody was added to warm KOSR medium at 1:10 dilution and incubated at 37 °C for 30 min. Wells were then washed twice with KOSR medium before medium containing secondary antibody, Goat anti Mouse Affinipure IgG+IgM (H + L) Alexafluor 647

(Jackson Laboratories) at 1:200 was added to each well, cells were incubated at 37 °C for 30 min. Wells were then washed once with KOSR medium and twice with FluoroBrite DMEM (ThermoFisher). After imaging the medium was replaced with fresh KOSR medium and returned to 37 °C incubator.

**Single cell cloning.** 96 well plates were coated with 0.1% gelatine and a layer of mouse embryonic feeders. Cells were harvested using accutase (ThermoFisher). After staining for flow cytometry, DAPI was added at 1:10,000 and used for live/dead discrimination. After gating on the BD Sortware programme (Supplementary Fig. S19c, d) the desired population was sorted as single cells directly into 96 well plates. The sort was indexed to retain information regarding the *MIXL1*-GFP and SSEA-3 expression levels. For single cell cloning the medium differed from standard culture. For this we used a 50/50 mix between standard KOSR medium and mTESR medium (Stem Cell Technologies) and the addition of 20 μM Synthechol (Sigma). During initial plating 10 μM of Rock Inhibitor (Y-27632, Generon) was added to the medium. Immediately after sorting into the wells the plates were centrifuged at 200 g for 1 min to aid attachment of the cells. After 2 days the medium was replaced with fresh medium to remove the ROCK inhibitor. Colonies were left to develop over 9–12 days before passaging the wells which looked to contained typical human PSC colonies. Colonies were passaged from 96 well plate into a 48 well plate by manual scrapping with a p200 tip, then aspirating and dispensing the dissected colony into one well of a 48 well plate. This 50/50 mix medium was used to grow clones until lines appeared to be growing stably, often up to the third passage before changing into standard KOSR medium.

**Embryoid bodies in neutral conditions.** To assess the trilineage potential, we used an approached which entailed the formation of Embryoid Bodies (EB) under neutral conditions, in this context neutral simply indicates that no exogenous cytokines or chemicals were added to guide differentiation. Cells were either used directly from flasks or after they had been FACS sorted for a particular population (Supplementary Fig. S19e). In either situation cells were resuspended in APEL 2 medium (Stem Cell Technologies) containing 10 μM of Rock Inhibitor (Y-27632, Generon) Cells were resuspended at 3000 cells per 50 μL. 50 μL of cells were added to non-adherent Grenier U bottom 96 well plate. After adding the cells plates were centrifuged at 200 g for 3 min, to pellet cells. Plates were incubated at 37 °C, 5%CO$_2$ for 7 days. After 7 days EBs were dissolved in Trizol and RNA extracted using the Norgen Total RNA Purification Kit. The RNA was converted to cDNA using the High Capacity cDNA Reverse Transcription Kit (Thermofisher). The samples were then loaded and run on the hPSC scorecards (ThermoFisher), results were analysed by ThermoFisher's scorecard software. Algorithm scores for EBs were normalised to the same cell line growing in E8 conditions.

**Directed embryoid body differentiation.** Differentiation protocols were adapted from Allison et al.[4]. Cells were prepared in the same manner as for the neutral EBs. APEL2 medium containing 10 μM of Rock Inhibitor (Y-27632, Generon) was supplemented with these reagents for each lineage; ectoderm (1 μM DMH1 (Tocris), 10 μMSB431542 (Tocris) and 100 ng/ml basic-FGF(Peprotech)), endoderm (100 ng/ml Activin-A (Peprotech), 1 ng/ml BMP4(Thermofisher) and mesoderm (20 ng/ml Activin-A (Peprotech), 20 ng/ml BMP4 (Thermofisher). EBs using the H9 T-Venus line were also supplemented with 1 μM IWP-2 (Tocris). EBs plates were placed in an incubator at 37 °C and 5%CO$_2$ for 10 days, then harvested and analysed by the hPSC scorecard method as per the neutral EBs.

**Neural crest differentiation.** Neural crest protocol was adapted from Hackland et al.[24]. Cells were harvested and plated at 40,000 cells/cm$^2$ on wells coated with Geltrex (Thermofisher) in neural crest medium (DMEM/F12 basal, Sigma, 1× N2 supplement, Thermofisher, 1× Non-essential Amino Acids, Thermofisher, 1× GlutaMax, Thermofisher, 2 μM SB431542, Tocris, 1 μM CHIR99021, Tocris, 1 μM DMH1, Tocris and BMP4 15 ng/ml, Gibco). Cells were plated either with 10 μM of Rock Inhibitor (Y-27632, Generon) or without Rock Inhibitor and 1 μM of IWP-2 (Tocris) for the first 48 h. Plates were placed in an incubator at 37 °C and 5%CO$_2$. Medium was replaced after 48 h with fresh neural crest medium. Medium was replaced every other day. Cells were fixed on the 5th day of neural crest differentiation and stained for SOX10 (Cell Signalling Technology, 89356S, 1:500).

**Blood differentiation.** The differentiation towards blood lineage precursors has been described previously[52,53]. Prior to the differentiation cells were split into flasks containing E8 or PRIMO Plus medium and left to grow for three days. OP9 stromal cells were grown in 10 cm dishes coated with 0.1% gelatine in OP9 growth medium (α-MEM (ThermoFisher), 10% Foetal Bovine serum (Thermofisher), 100 μM monothioglycerol (Thermofisher) and 100 U/mL of Penicillin–Streptomycin (Thermofisher)) until they had reached confluency. Human PSC were harvested as clumps by treatment with collagenase IV for 10–30 min and scraping from the flasks. A sister flask of both conditions was analysed for reporter gene and SSEA-3 expression by flow cytometry. Human PSC were seeded onto confluent layer of OP9 cells in OP9 growth medium. Medium is replaced fully (20 mL) on day 1, then half changes (10 mL) on day 4, 6 and 8. After 10 days, cells were harvested and stained with CD147 APC-conjugated antibody (R&D Systems, FAB3195A, 1:100), CD43 PE-conjugated antibody (Biolegend, 343203, 1:00) and

CD34 PECy7-conjugated antibody (Biolegend 343516, 1:160). Cells were analysed by flow cytometry and firstly gated on the pan-human marker CD147 to remove signal from remaining OP9 stromal cells.

**Differentiation time course.** Cells growing in E8 medium were supplemented with 3 μM CHIRON 99021 (Tocris) one day post passage. Cells were collected and stained with anti-SSEA-3 antibody at time points 6, 12, 18, 24, 48 and 72 h. Cells were sorted directly into Trizol. The medium was refreshed after 48 h with fresh E8 containing 3 μM CHIRON.

**Bulk sorting and RNA sequencing.** We sorted using a BD FACS Jazz (Supplementary Fig. S19f, g) or FACS Aria III (Supplementary Fig. S19a, b), we first began by sorting 10,000 cells from a given population into an eppendorf tube containing 200 μL DMEM/FCS. 1,000 cells are then reanalysed from this sorted fraction to ensure accurate sorting has taken place. Immediately after a successful reanalysis, 10,000 cells from the given population are sorted into an eppendorf tube containing 800 μL Trizol. Post sort, samples were sealed, vortexed and stored at −80 °C until extraction. Total RNA was extracted from the aqueous phase of thawed Trizol (Thermo Fisher) lysates after addition of 160 μl chloroform and precipitated by mixing an equal volume of isopropanol supplemented with 10ug of linear poly-acrylamide (Sigma). The RNA pellet was washed twice with 70% ethanol and resuspended in water. Total RNA was quality controlled and quantified using Bioanalyzer RNA 6000 Pico chip kit (Agilent). 1.5 ng total RNA was used with SMARTer.v3 or SMART-Seq.v4 kits (Clontech) to prepare cDNA libraries as per supplier instructions. cDNA libraries were purified with Agencourt AMPureXP (Beckmam) magnetic beads (at 0.8 bead to 1 cDNA ratio), washed twice with fresh 80% ethanol and eluted in EB buffer. Each of the cDNA libraries were quality checked using Fragment Analyzer High Sensitivity Large Fragment kit (Agilent), and quantified using Qubit High Sensitivity DNA kit (Thermo Fisher). 150 pg of each cDNA was used to prepare sequencing libraries with Nextera XT DNA Preparation Kit and Indexing primers (Illumina), but following a slightly modified version of the manufacturer's protocol involving fourfold volume miniaturisation throughout (12.5ul final volume). Nextera-XT indexed sequencing libraries were purified with Agencourt AMPureXP magnetic beads (at 0.8 bead to 1 cDNA ratio), washed twice with fresh 80% ethanol and eluted in EB buffer. Each of the indexed libraries were quality checked using Fragment Analyzer High Sensitivity Large Fragment kit, and quantified using Qubit High Sensitivity DNA kit. Indexed sequencing libraries were pooled to be equimolar. Library pools were quality checked using Bioanalyzer High Sensitivity DNA kit (Agilent), and quantified using Qubit High Sensitivity DNA kit before running out at 2pM on NextSeq High Output flow cells using 76-cycle paired-end sequencing of inserts. These indexed pools were demultiplexed through Illumina BaseSpace to provide pass-filter FASTQ output sample files. Raw fastq files from the RNA-seq experiments were run through a processing pipeline beginning with QC using FastQC v 0.11.5[54]. Following initial QC the reads were aligned using HISAT2 v2.1.0[55] to the human reference genome GRCh38[56]. Post-alignment QC and feature counts were performed using QoRTs v1.1.8[57] and feature quantification was done with cufflinks v 2.2.1[58]. Gene expression data was then analysed by Seqmonk v1.46[59] to produce heat maps and clustering. Differential gene expression analysis was performed using DESeq2[60] and Seqmonk intensity difference filter[59]. Gene Ontology enrichment was performed by ToppGene[61] and ReviGO[62].

**Single cell qPCR.** Samples were sorted using the flow cytometer, BD FacsAria III (Supplementary Fig. S19a, b). DAPI (Sigma) or Propidium Iodide (Sigma) was added for live/dead discrimination. Sorting population gates were set in the flow cytometry software, FACS DIVA, based on GFP expression (488 nm laser), and SSEA-3 (stained with Alexafluor 647 secondary antibody) expression (635 nm laser). Single cells from each selected substate were then directly deposited into 96 well plate with 4 μl of lysis buffer (65 μM of dNTP mix (Invitrogen 10297-018), 0.4% of NP40 (Sigma N-3516), 2.4 mM DTT (Invitrogen), 0.5 U/μl RNAseOUT (Invitrogen 10777-019) in nuclease-free water). cDNA synthesis and target specific pre-amplification was done using Cell Direct one-step qRT-PCR kit (Invitrogen 46-7201). Pre-amplification mastermix was added to each well; 6.25 μL of 2X Reaction Buffer, 1 μl of SuperScript III RT/Platinum Taq mix (Invitrogen 55549) and 1.5 μl of TaqMan assay mix. TaqMan assay mix was prepared by mixing equal volume of all target specific primers (Supplementary Table S1). No-RT controls were prepared with PlatinumTaq Polymerase (Invitrogen, 100021272) and no SuperScript III RT enzyme was included. The PCR conditions were: 60 min at 50 °C, 2 min at 95 °C and 25 cycles of 15 s at 95 °C and 4 min at 60 °C. Pre-amplified product was diluted 1–5 and loaded onto a 48.48 chip together with Taqman universal MasterMix (Applied Biosystems 4304437) and the Taqman assays listed in (Supplementary Table S1) with the appropriate loading reagents according to manufacturer's instructions (BioMark 48.48 Dynamic array platform (Fluidigm). Single cell qPCR analysis was performed by two pieces of software. Prior to analysis cells were screened and cells with particularly high CT values (>30) for the housekeeping genes *ACTB* and *RPS18* were removed, across all experiments 9 of 320 single cells were removed. We verified expression of genes with extra wells containing 10 cells rather than single cells, *MMP1* was removed from analysis, as it showed no expression in any samples. Depending on the application being used,

some pre-processing was performed. For Monocle2 software[18] and subsequent T-SNE plots, 999 values were set to 40, as this is the maximum number of cycles on the PCR machine. We used Genesis software[63] for heatmap production, where 999 values were removed and Z-scores were computed from all remaining values. For each gene, Z-scores were calculated using this formula: (Ct value – Ct mean)/standard deviation Cts.

**Reporting summary**. Further information on research design is available in the Nature Research Reporting Summary linked to this article.

## Data availability

RNA sequencing data that supports the findings of this study have been deposited in the ArrayExpress database at EMBL-EBI under the accession number E-MTAB-9474. All other supporting data are available from the corresponding author upon request. Source data are provided with this paper.

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

## Acknowledgements

This work was supported by grants from the European Community's Sixth and Seventh Framework Programs (LSHG-CT-2006-018739 and FP7/2007-2013-602423, PluriMes) and the MRC through the UK Regenerative Medicine Platform (grant no. MR/L012537/1 and MR/R015724/1). C.B. was supported by the Swedish Research Council (no. 2015-00135) and Marie Sklodowska Curie Actions, Cofund, Project INCA (no. 600398). C.J. was supported by Bloodwise. We thank Andrew Elefanty for providing HES3 and HES3-MIXL1 lines, Roger Pedersen and Daniel Ortmann for the providing H9 *T*-Venus line and Paul Gokhale and Anestis Tsakiridis for advice and insight. We also wish to acknowledge Jyoti Chhetri Bikram for technical support for the blood differentiation.

## Author contributions

P.W.A. and T.E. conceived the project. D.S. designed and performed the majority of experiments and analysis with aid from C.B., C.P., T.J.R.F., J.H., I.S.G., I.B., J.B., J.C. and C.J. D.S. wrote the manuscript with aid from P.W.A., T.E. and C.B. C.B. performed single cell qPCR reactions and J.B. and C.B. performed RNA sequencing, sequence alignment by C.J. Blood lineage differentiation and analysis was performed by J.W.

## Competing interests

The author, Dylan Stavish, is listed as the inventor on a patent submission (GB 1814084.8) pertaining to the formulation of the PRIMO media specified in the text. The other authors declare no competing interests.
