## [Peer Review File · Nature Communications]

Reviewers' comments:

Reviewer #1 (Remarks to the Author):

In this manuscript, the authors present evidence that human pluripotent stem cells can be “trapped” in an intermediate state between undifferentiated cells and mesodermal cells. To do so, they developed a culture condition that includes pro-differentiation factors, in particular a GSK3 inhibitor to activate Wnt signaling, and pro-pluripotency factors, such as LPA. Titrating the proper amounts of these compounds to produce a medium referred to as PRIMO (or PRIMO-plus) skewed the cells towards a mesodermal fate. They furthermore present data that this “substate” between pluripotent cells and mesodermally-fated cells is plastic, with cells being able to revert to an unbiased pluripotent state upon return to standard E8 medium.

Although the data presented here are overall convincing and appropriately interpreted, this study does not provide sufficient functional analysis of these substate cells. For example, the authors should provide a more thorough characterization of these cells, especially after prolonged culture in PRIMO conditions (after 10+ passages), and then again after they have been returned to basal culture conditions. It is currently unclear how stable this substate is. And can cells be expanded to large amounts in this condition so that they may be of utility for future cell- and tissue-engineering approaches. Such experiments are needed to address the stability and utility of the “trapped MIXL1(+)/SSEA4(+)” state in PRIMO medium.

Functional characterization, at minimum, should include a measure of proliferative potential, expression of pluripotency markers, cell morphology, differentiation potential (embryoid body and/or teratoma formation), and assessment of genomic stability (euploid or aneuploid). The current characterization relies almost entirely on gene expression, which is insufficient to make the argument that these cells are in fact a stabilized, or trapped, mesodermal stem or progenitor population.

The authors also need to demonstrate that these cells harbor true lineage bias by showing efficient differentiation into more mature mesodermal derivatives, such as muscle, blood, urinary tract, etc. It would also be interesting to know whether these trapped cells are biased towards a specific subset of mesoderm, including lateral plate, paraxial or intermediate mesoderm. Are these cells capable of differentiating into any mesodermal derivative, or are they restricted to one subset? In addition, are these cells unable to differentiate into endo- and ecto-dermal derivatives? Expression of genes associated with each of these germ layers, as shown in Figures 5 and 6, is a good start but more is needed; for example, can these cells be forced to differentiate into gut tube derivatives (pancreas, liver or lung) or into neural progenitors? A thorough analysis of the differentiation potential of these cells is essential to establish that these cells are trapped in a mesodermal state.

A few minor points:

Figure 1: for the RNA-seq analysis of the 3 bulk-sorted cell populations, the authors should show RPKMs for MIXL1 (or GFP) to confirm that MIXL1 transcript levels correlate with protein levels.

Line 59: briefly describe the MIXL1-GFP line and include original reference (currently Ref #11) here. Does insertion of GFP disrupt the MIXL1 gene? If so, is there any haploinsufficiency?

Line 123-124: Figure 2b is an image of a single colony, not six clonal lines as the text suggests.

Several prior studies have used pluripotent stem cells to establish lineage restricted cell populations, including mesoderm and endoderm. Such work, including the following references, should be included and discussed:

- Bakre et al. Generation of multipotential mesendodermal progenitors from mouse embryonic stem cells via sustained Wnt pathway activation. J Biol Chem. 2007
- Cheng et al. Self-renewing endodermal progenitor lines generated from human pluripotent stem cells. Cell Stem Cell. 2012
- Kumar et al. Generation of an expandable intermediate mesoderm restricted progenitor cell line from human pluripotent stem cells. Elife. 2015
- Wang et al. Targeting SOX17 in human embryonic stem cells creates unique strategies for isolating and analyzing developing endoderm. Cell Stem Cell. 2011.

Reviewer #2 (Remarks to the Author):

The manuscript from Stavish and colleagues reports that human Pluripotent Stem Cells can be "trapped" in a Mesoderm biased state by combining a GSK3 inhibitor and LPA. The manuscript is well written and the results are very interesting.

I would suggest to clarify the following two points:

1) the MIXL1-GFP reporter strategy is based on detection of GFP, which is a rather stable protein. Therefore some of the MIXL1-GFP positive cells might in fact be cells that do not express endogenous MIXL1 transcripts at the time of analysis.

This seems to be the case if we look at Figure 1d and Supplementary Figure 4b, where a fraction of MIXL1-GFP positive cells do not express MIXL1 by single-cell qPCR.

I would suggest to repeat some of the analyses considering the actual MIXL1 expression state measured by single-cell qPCR. For example, in Figure 1c, some of the MIXL1(+)/SSEA3(+) cells cluster very close to MIXL1(-)/SSEA3(+), are they MIXL1 negative?

Perhaps colouring the dots in figure 1c according to MIXL1 expression would be more informative.

Also In figure 1d some of the MIXL1(+)/SSEA3(+) cells show no expression of MIXL1 and robust expression of SOX2, NANOG and OCT4, while those expressing MIXL1 show lower levels of pluripotency markers and higher levels of T, as also shows in figure 1e.

In other words, are the "true" MIXL1(+)/SSEA3(+) cells so heterogeneous, or is the reporter strategy generating some technical noise which makes the results less clear?

2) A key conclusion is that PRIMO MIXL1(+)/SSEA3(+) are pluripotent and biased towards Mesoderm. It is very clear that such cells readily form Mesoderm during EB differentiation, more efficiently - or more rapidly - than hPSCs cultured in E8.

However, I am less convinced about their pluripotency.

PRIMO MIXL1(+)/SSEA3(+) cells can revert to the pluripotent state, but this happens in ~10% of clones (Figure 5b) and requires multiple passages in different media, indicating that they can acquire pluripotency, not that they are pluripotent.

Indeed, figure 5d shows that PRIMO cells are nearly unable to activate Ectoderm markers.

Are PRIMO cells able to form Ectoderm, Mesoderm and Endoderm via monolayer differentiation? (e.g. dual Smad inhibition protocol for Neuroectoderm).

If PRIMO cells are unable to form Neuroectoderm it would be more fair to define them as stable Mesoderm progenitors, with the capacity to revert to the pluripotent state.

This would be equally interesting but more accurate.

Response to Reviewers' Comments

Note: **Highlights in Yellow** are indicating specific changes to manuscript

Reviewer #1

1. *Although the data presented here are overall convincing and appropriately interpreted, this study does not provide sufficient functional analysis of these substate cells. For example, the authors should provide a more thorough characterization of these cells, especially after prolonged culture in PRIMO conditions (after 10+ passages), and then again after they have been returned to basal culture conditions. It is currently unclear how stable this substate is. And can cells be expanded to large amounts in this condition so that they may be of utility for future cell- and tissue-engineering approaches. Such experiments are needed to address the stability and utility of the “trapped MIXL1(+)/SSEA4(+)” state in PRIMO medium.*

We appreciate the reviewer's note that our data are 'overall convincing and appropriately interpreted' and particularly for pointing out the importance of the long term stability of the mesoderm biased substates if they are to be scaled up for practical application. Consequently, we have reviewed the extent to which prolonged culture of such cells in PRIMO may be necessary for their general utility. In fact, given the efficiency of the growth of the cells that we have observed in PRIMO, we estimate that it would be possible to generate more than 10^{10} cells after even only three passages, and up to 10^{17} after ten passages, from an initial seeding of one million cells, from E8 into PRIMO. Given the general concern about the acquisition of mutations in hPSC after extended culture (e.g. see the discussion of the ISCI project –*Andrews et al 2017 Stem Cell Reports 9:1-4*), it is always desirable to keep levels of expansion to the minimum required, so we anticipate that applications of the mesoderm biased cells will generally require substantially fewer than ten passages in PRIMO medium. To emphasise the stability of the mesoderm biased substate over such periods of culture we have now supplemented the data already included in the manuscript with further data for another cell line (H9 T-Venus), to confirm the robustness of maintaining hPSC in the mesoderm-biased substate for at least ten passages. (see, supplementary Figures 11b, 12 and 13 and lines 336- 344 of the manuscript). We have also added commentary in the Discussion about the potential for scale up and utility of mesoderm biased cells from culture in PRIMO (see text: lines 442-450).

2. *Functional characterization, at minimum, should include a measure of proliferative potential, expression of pluripotency markers, cell morphology, differentiation potential (embryoid body and/or teratoma formation), and assessment of genomic stability (euploid or aneuploid). The current characterization relies almost entirely on gene expression, which is insufficient to make the argument that these cells are in fact a stabilized, or trapped, mesodermal stem or progenitor population.*

To provide further functional characterisation of the mesoderm biased cells, and to supplement the neutral EB assays of the same cells already included in the manuscript, we have now carried out directed EB differentiation of HES3 MIXL1 and H9-T-Venus cells grown in PRIMO medium. These data show that pluripotency is retained in PRIMO medium since these cells could generate both ectoderm and

endoderm as well as mesoderm, when forced in these directions, in contrast to the strong bias to mesoderm differentiation in EBs under neutral conditions. This conclusion of pluripotency of PRIMO cultured cells was also shown by our ability to induce their differentiation to neural crest derivatives, often termed the 'fourth germ layer', from three cell lines. The results of these experiments are now shown in Fig. 5f-g and supplementary Fig. 9, and discussed in the manuscript – lines 292-306. (See also our response to Comments 1.4 and 2.2 below).

We have now confirmed the genetic stability of the cells in PRIMO by including G-banding karyotypes from 30 metaphase spreads for HES3 MIXL1 and H9 T-Venus at passage 10 which revealed no karyotypic abnormalities (See Supplementary Fig 13 and lines 342-344 in the manuscript). Further, as requested, we have now provided additional characterisation by including data on proliferation, cell morphology (see Supplementary Fig. 15) and expression of pluripotency associated markers (see Supplementary Figures 11, 12 and 14).

3. The authors also need to demonstrate that these cells harbor true lineage bias by showing efficient differentiation into more mature mesodermal derivatives, such as muscle, blood, urinary tract, etc.

We appreciate the reviewer's question, which certainly pertinent to the utility of the PRIMO medium. To address this, it is necessary to use a protocol that is robust and reproducible though, ideally, not especially efficient. We have, accordingly, focused upon differentiation towards a blood lineage utilising a well-established yet not overly efficient protocol in which hPSC are plated on OP9 stroma cells and allowed to differentiate until day 10. When compared to hPSC maintained in E8 cultures, cells maintained in PRIMO showed between a 5-15% increase in CD43+ cells. Further, cells from PRIMO cultures generated between 2-3 million more CD43+ cells compare to cells from E8 cultures, representing a ~30-40% increase in yield of haematopoietic cells. Here we calculated yield using the starting numbers seeded for differentiation. However, overall, the yield is actually greater since we observed substantially better growth of cells in PRIMO than in E8, prior to differentiation (PRIMO cultures had on average 1.4x as many cells as their E8 counterparts).

These results are presented in new Figure 5e and referred to in the manuscript, lines 279 to 291.

4. It would also be interesting to know whether these trapped cells are biased towards a specific subset of mesoderm, including lateral plate, paraxial or intermediate mesoderm. Are these cells capable of differentiating into any mesodermal derivative, or are they restricted to one subset? In addition, are these cells unable to differentiate into endo- and ecto-dermal derivatives? Expression of genes associated with each of these germ layers, as shown in Figures 5 and 6, is a good start but more is needed; for example, can these cells be forced to differentiate into gut tube derivatives (pancreas, liver or lung) or into neural progenitors? A thorough analysis of the differentiation potential of these cells is essential to establish that these cells are trapped in a mesodermal state.

We are grateful to the reviewer for raising these interesting questions. However, as discussed above, in response to point 1.2, we have now included new data to confirm that the mesoderm biased cells trapped by culture in PRIMO still retain full pluripotency. Thus, when EBs produced by these cells were cultured under conditions designed to force differentiation towards one of the three germ layers, endoderm predominated under endoderm conditions, and ectoderm predominated under ectoderm conditions, in contrast to EBs produced under the neutral conditions, when mesoderm differentiation predominated. These functional data imply that the

mesoderm biased cells in PRIMO still correspond to a subset of cells within the undifferentiated stem cell compartment. This is also consistent with other characteristics of these cells, such as their continued expression of pluripotency associated markers, and with our time course data indicating that the cells represent a very early stage in the mesoderm trajectory and so most likely to a stage before specification of particular subtypes of mesoderm. Consequently, we think it improbable that the cells in PRIMO will exhibit commitment for particular subtypes of mesodermal derivatives. We have now presented these new data in Fig. 5f, and included a discussion of these ideas – see lines 422-432.

Minor points:

i) Figure 1: for the RNA-seq analysis of the 3 bulk-sorted cell populations, the authors should show RPKMs for MIXL1 (or GFP) to confirm that MIXL1 transcript levels correlate with protein levels.

The FPKM values for MIXL1 have now been added to Supplementary Figure 1 and in the manuscript, line 76 -77, we have now added the sentence: “The expression of MIXL1-GFP appeared correlate well with the MIXL1 gene expression in the RNAseq data (Supplementary Fig. 1a)”

ii) Line 59: briefly describe the MIXL1-GFP line and include original reference (currently Ref #11) here. Does insertion of GFP disrupt the MIXL1 gene? If so, is there any haploinsufficiency?

We have now added (Line 59-64) a more detailed explanation of the cell line and reference, as well as citations of publications that have successfully used the line for production of terminal differentiated cells.

iii) Line 123-124: Figure 2b is an image of a single colony, not six clonal lines as the text suggests.

This was an error in the text and should refer to Figure 2a which relates to the original sorting position of the six clonal lines taken forward; it has now been amended in the manuscript.

iv) Several prior studies have used pluripotent stem cells to establish lineage restricted cell populations, including mesoderm and endoderm. Such work, including the following references, should be included and discussed:

- Bakre et al. Generation of multipotential mesendodermal progenitors from mouse embryonic stem cells via sustained Wnt pathway activation. *J Biol Chem.* 2007
- Cheng et al. Self-renewing endodermal progenitor lines generated from human pluripotent stem cells. *Cell Stem Cell.* 2012
- Kumar et al. Generation of an expandable intermediate mesoderm restricted progenitor cell line from human pluripotent stem cells. *Elife.* 2015
- Wang et al. Targeting SOX17 in human embryonic stem cells creates unique strategies for isolating and analyzing developing endoderm. *Cell Stem Cell.* 2011.

We have now included new passage in the Discussion (lines 422-432) in which we have incorporated and discussed these references, which highlight not only similarities with our work of trapping a substate but also the differences of our system which can retain cells with a pluripotent rather than multipotent potential:

"While the generation or identification of multipotent progenitors towards mesoderm and endoderm from PSC has been demonstrated previously in both mouse³² and human³³⁻³⁵ PSC in vitro, these studies indicated a lack of traditional pluripotent markers such as NANOG, POU5F1 and SOX2 and a lineage restricted differentiation potential, representing trapping of later stages of differentiation. An important, distinction is that this substate we describe here was plastic with cells being able to further differentiate or, when returned to E8 medium, revert to an unbiased stem cell state."

Reviewer #2

1. *the MIXL1-GFP reporter strategy is based on detection of GFP, which is a rather stable protein. Therefore some of the MIXL1-GFP positive cells might in fact be cells that do not express endogenous MIXL1 transcripts at the time of analysis. This seems to be the case if we look at Figure 1d and Supplementary Figure 4b, where a fraction of MIXL1-GFP positive cells do not express MIXL1 by single-cell qPCR. I would suggest to repeat some of the analyses considering the actual MIXL1 expression state measured by single-cell qPCR. For example, in Figure 1c, some of the MIXL1(+)/SSEA3(+) cells cluster very close to MIXL1(-)/SSEA3(+), are they MIXL1 negative? Perhaps colouring the dots in figure 1c according to MIXL1 expression would be more informative. Also in figure 1d some of the MIXL1(+)/SSEA3(+) cells show no expression of MIXL1 and robust expression of SOX2, NANOG and OCT4, while those expressing MIXL1 show lower levels of pluripotency markers and higher levels of T, as also shows in figure 1e. In other words, are the "true" MIXL1(+)/SSEA3(+) cells so heterogenous, or is the reporter strategy generating some technical noise which makes the results less clear?*

We agree that one of the problems with the reporter strategy is that persistence of the reporter protein may have led to some technical noise in our system. To clarify this point, we have recoloured the TSNE plot, based on MIXL1 intensity (Supplementary Fig 1b). Because of the disparity between the actual MIXL1 RNA levels and MIXL1-GFP expression levels, for clarity we have decided to refer to cells identified by FACS as MIXL1-GFP(+). On the other hand, we have also added a pseudotime analysis based on single cell gene expression data (see Supplementary Fig 1c-d and lines 106-117 in the manuscript), which shows that cells segregated by MIXL1-GFP/SSEA-3 do fit well on a differentiation trajectory suggesting that noise associated with the use of the reporter has not significantly affected our conclusions. Because of the disparity between the actual MIXL1 RNA levels and MIXL1-GFP expression levels for clarity we have decided to refer to cells identified by FACS as MIXL1-GFP(+) instead.

2. *A key conclusion is that PRIMO MIXL1(+)/SSEA3(+) are pluripotent and biased towards Mesoderm. It is very clear that such cells readily form Mesoderm during EB differentiation, more efficiently - or more rapidly - than hPSCs cultured in E8. However, I am less convinced about their pluripotency. PRIMO MIXL1(+)/SSEA3(+) cells can revert to the pluripotent state, but this happens in ~10% of clones (Figure 5b) and requires multiple passages in different media, indicating that they can acquire pluripotency, not that they are pluripotent. Indeed, figure 5d shows that PRIMO cells are nearly unable to activate Ectoderm markers. Are PRIMO cells able to form Ectoderm, Mesoderm and Endoderm via monolayer differentiation? (e.g. dual Smad inhibition protocol for Neuroectoderm). If PRIMO cells are unable to form Neuroectoderm it would be more fair to define them as stable Mesoderm progenitors, with the capacity to revert to the pluripotent state. This would be equally interesting but more accurate.*

The distinction between whether the cells in PRIMO exhibit a truly pluripotent, if biased, substate or a mesoderm committed state that is subject to a relatively low frequency of reversion to a pluripotent state, is an important question that speaks to our views of the structure of the pluripotent state and how cells exit this state. Unfortunately, the

low cloning efficiency of hPSC makes it difficult to distinguish the two hypotheses based upon the recovery of clonal lines. However, we have now grown both HES3-MIXL1 and H9 T-Venus for 3 passages in PRIMO medium and have compared “neutral” EBs from these cells, and from cells transitioned into E8 medium for just one passage: Whereas the EBs direct from PRIMO showed a strong mesoderm lineage bias, those from cells transitioned into E8 medium formed unbiased EBs containing derivatives of all germ layers. In addition, as also described in our responses to Reviewer 1.2 and 1.4, we found that when cells from PRIMO culture form EBs under conditions designed to specifically to promote ectoderm or endoderm, then differentiation in those directions predominated, in contrast to the mesoderm bias under neutral differentiation conditions. Further, cells from PRIMO could be induced readily to form neural crest. Taken together these data are consistent with our view that the mesoderm biased cells from culture in PRIMO represent a mesoderm biased substate of pluripotency rather than a mesoderm progenitor state. We have now included these data in the revised manuscript (see Fig. 5f-g and supplementary Fig. 9), and summarised our argument in the Discussion (lines 422-432).

REVIEWERS' COMMENTS:

Reviewer #1 (Remarks to the Author):

The authors have satisfactorily addressed all of my concerns and critiques raised in my first review. I only have one follow-up comment on the need to thoroughly address publications that precede this work. Specifically, in regards to the paper by Bakre et al. (2007), the authors should note that the mesendodermal progenitor cells (MPC) described in this work revert upon withdrawal of GSK3 inhibitor, the compound that drives cells to this state. Therefore, the cells produced by Bakre et al. may in fact be more closely related to the cells described in this current work than the authors appreciate and acknowledge, and a more thorough discussion is warranted.

Karl Willert, Professor, UCSD

Reviewer #2 (Remarks to the Author):

The authors addressed all the points raised by the referees in a satisfactory way. There are some minor mistakes (e.g. labels of Fig. S1c are wrong) so I would suggest to check carefully the manuscript and figures. Overall the paper is now much more robust therefore I fully support its publication.

Graziano Martello

Response to Reviewers' Comments

Note: **Highlights in Yellow** are indicating specific changes to manuscript

Reviewer #1 (Remarks to the Author):

1. The authors have satisfactorily addressed all of my concerns and critiques raised in my first review. I only have one follow-up comment on the need to thoroughly address publications that precede this work. Specifically, in regards to the paper by Bakre et al. (2007), the authors should note that the mesendodermal progenitor cells (MPC) described in this work revert upon withdrawal of GSK3 inhibitor, the compound that drives cells to this state. Therefore, the cells produced by Bakre et al. may in fact be more closely related to the cells described in this current work than the authors appreciate and acknowledge, and a more thorough discussion is warranted.

Karl Willert, Professor, UCSD

We are happy that our revisions have addressed your concerns. You have highlighted the similarity with our cells and the ones described by Bakre et al, 2007. We agree with your sentiments that our state and what they describe in mouse might be similar, albeit with a slower reversion time for theirs. **We have therefore added the following passage to the discussion (see text lines 423-426):**

"Bakre et al, 2007, described a seemingly similar mesoderm biased state in mouse PSC induced into a mesoderm biased state through manipulation of WNT signalling³⁴. However, reversion of their cells to an unbiased state took two weeks of culturing."

Reviewer #2 (Remarks to the Author):

The authors addressed all the points raised by the referees in a satisfactory way. There are some minor mistakes (e.g. labels of Fig. S1c are wrong) so I would suggest to check carefully the manuscript and figures.

Overall the paper is now much more robust therefore I fully support its publication.

Graziano Martello

We are glad that our corrections proved satisfactory. **The labels on figure S1c have been amended** and the figures have now been checked for any other errors.